# Sensitivity-Aware Amortized Bayesian Inference

**Lasse Elsemüller**                                         *lasse.elsemueller@gmail.com*
*Heidelberg University*

**Hans Olischläger**
*Heidelberg University*

**Marvin Schmitt**
*University of Stuttgart*

**Paul-Christian Bürkner**
*TU Dortmund University*

**Ullrich Köthe**
*Heidelberg University*

**Stefan T. Radev**                                          *stefan.radev93@gmail.com*
*Rensselaer Polytechnic Institute*

**Reviewed on OpenReview:** *https://openreview.net/forum?id=Kxtpa9rvM0*

## Abstract

Sensitivity analyses reveal the influence of various modeling choices on the outcomes of statistical analyses. While theoretically appealing, they are overwhelmingly inefficient for complex Bayesian models. In this work, we propose sensitivity-aware amortized Bayesian inference (SA-ABI), a multifaceted approach to efficiently integrate sensitivity analyses into simulation-based inference with neural networks. First, we utilize weight sharing to encode the structural similarities between alternative likelihood and prior specifications in the training process with minimal computational overhead. Second, we leverage the rapid inference of neural networks to assess sensitivity to data perturbations and preprocessing steps. In contrast to most other Bayesian approaches, both steps circumvent the costly bottleneck of refitting the model for each choice of likelihood, prior, or data set. Finally, we propose to use deep ensembles to detect sensitivity arising from unreliable approximation (e.g., due to model misspecification). We demonstrate the effectiveness of our method in applied modeling problems, ranging from disease outbreak dynamics and global warming thresholds to human decision-making. Our results support sensitivity-aware inference as a default choice for amortized Bayesian workflows, automatically providing modelers with insights into otherwise hidden dimensions.

## 1 Introduction

Statistical inference aims to extract meaningful insights from empirical data through a series of analytical procedures. Acknowledging that each of these procedures involves a myriad of implicit choices and assumptions, *any single analysis hides an iceberg of uncertainty* (Wagenmakers et al., 2022). We consider *sensitivity analysis* as a formal approach to shed light on this very iceberg of uncertainty.

For instance, global warming forecasts can change depending on the assumed earth system model. In other words: Climate change analyses can be sensitive to the underlying observation model (i.e., *likelihood*; see **Experiment 2**). Yet, the likelihood is not the only model component that can induce sensitivity. The prior assumptions, the approximation algorithm, and the specifics of the collected data contribute further uncertainty to the results (Bürkner et al., 2022).

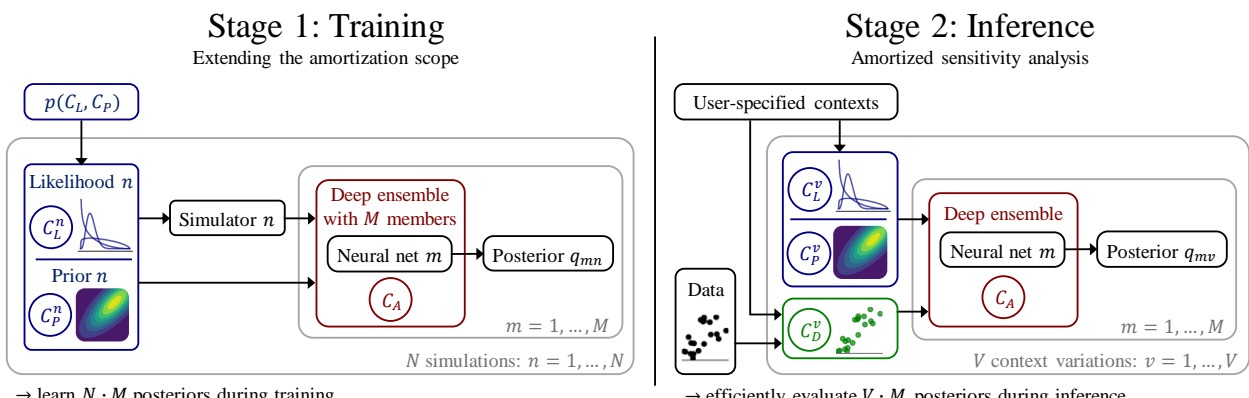

Figure 1: Our proposed approach for sensitivity-aware amortized Bayesian inference (SA-ABI). **Stage 1:** During training, a distribution $p(C_L, C_P)$ over plausible likelihood and prior choices is encoded via context variables $C_L$ and $C_P$ in a deep ensemble of neural approximators. **Stage 2:** During inference, we cast costly model refits as a near-instant neural network prediction task conditioned on user-specified context $C$. Our amortized neural approach unlocks fast large-scale sensitivity analyses of all components in a Bayesian model: likelihood ($C_L$), prior ($C_P$), data ($C_D$), and approximator ($C_A$). **Experiment 3** uses $V = 8\,100$ variations in prior and data alongside $M = 20$ deep ensemble members. The resulting amortized sensitivity analysis encompassing $V \cdot M = 162\,000$ approximate posteriors would have been infeasible with existing methods.

Classical sensitivity analyses rely on costly model refitting under each configuration and quickly become infeasible for both likelihood-based (e.g., MCMC; Neal, 2011) and simulation-based inference (SBI, Cranmer et al., 2020).

Recently, SBI methods have been accelerated through amortized Bayesian inference (ABI; Radev et al., 2020; Gonçalves et al., 2020; Avecilla et al., 2022), where neural networks learn probabilistic inference tasks and compensate for the training effort with rapid inference on many unseen data sets. As we demonstrate in this paper, this directly enables a large-scale assessment of data sensitivity. However, assessing likelihood, prior, and approximator sensitivity remains extremely challenging in standard ABI applications, which may be costly to train even for a single configuration. To address this gap, we investigate *sensitivity-aware amortized Bayesian inference* (SA-ABI) and demonstrate that it unlocks highly efficient and multifaceted sensitivity analyses in realistic ABI applications (cf. Figure 1). Our main contributions are:

1. We conceptualize sensitivity via implicit context variables and integrate established methods for sensitivity analysis into amortized Bayesian inference;

2. We investigate a context-aware neural architecture to quantify likelihood and prior sensitivity at inference time with minimal computational overhead and no notable loss of accuracy;

3. We assess approximator sensitivity via deep ensembles and data sensitivity via the near-instant ABI inference;

4. We demonstrate the utility of SA-ABI for Bayesian parameter estimation as well as Bayesian model comparison in three real-world scenarios of scientific interest, investigating sensitivity under up to $162\,000$ configurations.

## 2 Background

What follows is a brief overview of Bayesian parameter estimation, model comparison, amortized Bayesian inference, and learnable summary statistics. Readers familiar with these topics can safely jump directly to **Section 3**.

### 2.1 Bayesian Inference

**Bayesian Parameter Estimation**    In Bayesian parameter estimation, the key quantity is the *posterior distribution*,

$$p(\boldsymbol{\theta} \,|\, \boldsymbol{x}) = \frac{p(\boldsymbol{x} \,|\, \boldsymbol{\theta})p(\boldsymbol{\theta})}{p(\boldsymbol{x})}, \tag{1}$$

which combines the likelihood $p(\boldsymbol{x} \mid \boldsymbol{\theta})$ with prior information about parameter distributions $p(\boldsymbol{\theta})$, normalized by the (analytically intractable) marginal likelihood $p(\boldsymbol{x})$.

**Bayesian Model Comparison**    In many scientific applications, no single generative model can provide *the* ultimate explanation for a data set $\boldsymbol{x}$. Instead, a set of models $\mathcal{M} = \{\mathcal{M}_1, \mathcal{M}_2, \ldots, \mathcal{M}_J\}$ is plausible. Bayesian model comparison aims to find the "best" model within $\mathcal{M}$. In (prior-predictive) Bayesian model comparison, a model's *marginal likelihood* $p(\boldsymbol{x} \mid \mathcal{M}_j)$ now takes the central role:

$$p(\boldsymbol{x} \mid \mathcal{M}_j) = \int p(\boldsymbol{x} \mid \boldsymbol{\theta}, \mathcal{M}_j)\, p(\boldsymbol{\theta} \mid \mathcal{M}_j)\, \mathrm{d}\boldsymbol{\theta}. \tag{2}$$

Marginalizing the likelihood over the parameter space automatically encodes Occam's razor through a preference for models with limited prior predictive flexibility (MacKay, 2003). The *posterior model probabilities* of competing models can be computed as

$$p(\mathcal{M}_j \mid \boldsymbol{x}) = \frac{p(\boldsymbol{x} \mid \mathcal{M}_j)\, p(\mathcal{M}_j)}{\sum_{\mathcal{M}} p(\boldsymbol{x} \mid \mathcal{M})\, p(\mathcal{M})}, \tag{3}$$

where $p(\mathcal{M})$ is the prior distribution over the model space.

## 2.2  Simulation-Based Inference

Both Bayesian parameter estimation and model comparison have traditionally been limited by the ability to efficiently evaluate a model's likelihood density $p(\boldsymbol{x} \mid \boldsymbol{\theta})$. Likelihood-based methods (e.g., MCMC) assume that the distributional family of the likelihood is explicitly known and can be evaluated analytically or numerically for any pair $(\boldsymbol{x}, \boldsymbol{\theta})$. Differently, simulation-based approaches only require simulations from a simulation program $G$,

$$\boldsymbol{x} = G(\boldsymbol{\theta}, \boldsymbol{\xi}) \quad \text{with} \quad \boldsymbol{\xi} \sim p(\boldsymbol{\xi} \mid \boldsymbol{\theta}), \boldsymbol{\theta} \sim p(\boldsymbol{\theta}), \tag{4}$$

with latent program states or "outsourced" noise $\boldsymbol{\xi}$. A single execution of such a program corresponds to generating samples from the Bayesian joint model $(\boldsymbol{\theta}, \boldsymbol{x}) \sim p(\boldsymbol{\theta}, \boldsymbol{x})$, since the execution paths of the simulation program define an *implicit likelihood* (e.g., Cranmer et al., 2020; Diggle & Gratton, 1984; Marin et al., 2012)

$$p(\boldsymbol{x} \mid \boldsymbol{\theta}) = \int \delta(\boldsymbol{x} - G(\boldsymbol{\theta}, \boldsymbol{\xi}))\, p(\boldsymbol{\xi} \mid \boldsymbol{\theta})\, \mathrm{d}\boldsymbol{\xi}, \tag{5}$$

where $\delta$ is the Dirac delta function. However, the above equation (5) is analytically intractable for any simulation program of practical interest, turning simulation-based inference into a computational challenge.

## 2.3  Amortized Bayesian Inference

Amortized Bayesian inference (ABI) leverages simulations from $G$ to solve Bayesian inference tasks with neural networks in real time after an initial training phase. To achieve this, neural networks learn to encode the relationship between simulated data and model states during training. As a result, costly probabilistic inference is replaced with a neural network prediction task. For parameter estimation, generative neural networks act as conditional neural density approximators of the parameter posterior $p(\boldsymbol{\theta} \mid \boldsymbol{x})$ (Greenberg et al., 2019; Radev et al., 2020). Model comparison, on the other hand, can be framed as a probabilistic classification problem which is addressed with discriminative neural networks to approximate posterior model probabilities $p(\mathcal{M} \mid \boldsymbol{x})$ (Pudlo et al., 2016; Radev et al., 2021a).

## 2.4  End-to-end Summary Statistics

Common neural density estimators hinge on fixed-length vector-valued inputs – a requirement that is violated by widespread data formats such as sets of $i.i.d.$ observations or time series. To this end, previous research explored ways to learn end-to-end summary statistics for flexible adaption to the individual data structure and inference task (Chan et al., 2018; Wiqvist et al., 2019; Radev et al., 2020; Chen et al., 2020; 2023). In a nutshell, a summary network $h_\psi$ compresses input data $\boldsymbol{x}$ of variable size to a fixed-length vector of learned summary statistics $h_\psi(\boldsymbol{x})$ by exploiting probabilistic symmetries in the data (e.g., permutation-invariant networks for exchangeable data; Bloem-Reddy & Teh, 2020). The resulting *embedding* is fed to an inference network $f_\phi$ that approximates the posterior (e.g., with a conditional normalizing flow). The summary network $h_\psi$ and the inference network $f_\phi$ are simultaneously trained end-to-end in order to learn optimal summary statistics for the inference task.

## 3 Methods

### 3.1 Extending the Amortization Scope

**Sensitivity Sources as Context Variables**  The PAD framework (Bürkner et al., 2022) defines a Bayesian model as a combination of the joint **P**robability distribution $p(\boldsymbol{\theta}, \boldsymbol{x} \mid \mathcal{M})$, which can be factorized into likelihood $p(\boldsymbol{x} \mid \boldsymbol{\theta}, \mathcal{M})$ and prior $p(\boldsymbol{\theta} \mid \mathcal{M})$, the posterior **A**pproximator, and the observed **D**ata $\boldsymbol{x}_{\text{obs}}$. Building on this decomposition, we define *sensitivity* to a component of a Bayesian model as the extent of change in inferential results induced by perturbations in any of these components (Bürkner et al., 2022). We will refer to the opposite of sensitivity as *robustness*: When an inference procedure is robust, it is not sensitive to changes in model components.

To enable systematic and comprehensive investigations of sensitivity, we consider sources of perturbations as context variables that implicitly shape inferential results (see Figure 1) and can be cheaply varied as conditioning variables (or hyperparameters) in our simulation-based approach. In contrast, standard Bayesian workflows treat context variables as fixed and thus cannot typically investigate their effect explicitly without re-doing the entire analysis.

In the following, we denote context variables as $C_L$ (likelihood), $C_P$ (prior), $C_A$ (approximator), $C_D$ (data), and refer to the entirety of context variables as $C = (C_L, C_P, C_A, C_D)$. Accordingly, we refer to posteriors with explicitly encoded context variables as $p(\boldsymbol{\theta} \mid \boldsymbol{x}, C)$ for parameter estimation and $p(\mathcal{M}_j \mid \boldsymbol{x}, C)$ for model comparison.

**Sensitivity-Aware Training**  Before discussing key facets of practical sensitivity analysis, we describe extensions to standard simulation-based training that allow these analyses to be performed *efficiently*. Specifically, we eliminate the necessity of retraining neural approximators for every aspect of a sensitivity analysis by incorporating the **T**raining contexts $C_T = (C_L, C_P)$ into the network's amortization scope (Radev, 2021). Since we realize $C_A$ via deep ensembles and $C_D$ only during inference (see **Section 3.2**), neither of these contexts influences the training objective of single ensemble members.

For sensitivity-aware parameter estimation (PE), we incorporate the context $C_T$ into the standard negative log-posterior objective via

$$\mathcal{L}^{\text{PE}}(\boldsymbol{\phi}, \boldsymbol{\psi}; C_T) = \mathbb{E}\Big[ - \log q_{\boldsymbol{\phi}}(\boldsymbol{\theta} \mid h_{\boldsymbol{\psi}}(\boldsymbol{x}), C_T)\Big]. \tag{6}$$

The expectation $\mathbb{E}$ is here taken over a contextualized joint Bayesian model $p(\boldsymbol{\theta}, \boldsymbol{x} \mid C_T)$, which produces tuples of training parameters and corresponding data $(\boldsymbol{\theta}, \boldsymbol{x})$ for the given context $C_T$.

Analogously, for sensitivity-aware Bayesian model comparison (BMC), we can target the approximate posterior model probability $q_{\boldsymbol{\phi}}(\mathcal{M} \mid h_{\boldsymbol{\psi}}(\boldsymbol{x}), C_T)$ via the cross-entropy

$$\mathcal{L}^{\text{BMC}}(\boldsymbol{\phi}, \boldsymbol{\psi}; C_T) = \mathbb{E}\Big[ - \sum_{j=1}^{J} \mathbb{I}_{\mathcal{M}_j} \log q_{\boldsymbol{\phi}}(\mathcal{M}_j \mid h_{\boldsymbol{\psi}}(\boldsymbol{x}), C_T)\Big], \tag{7}$$

where the expectation $\mathbb{E}$ is taken with respect to a contextualized generative mixture of Bayesian models $p(\mathcal{M}_j \mid C_T) \, p(\boldsymbol{x} \mid \mathcal{M}_j, C_T)$ producing tuples of model indices and associated simulated data $(\mathcal{M}_j, \boldsymbol{x})$. The indicator function $\mathbb{I}_{\mathcal{M}_j}$ denotes a one-hot encoding for the true model index, i.e., $\mathbb{I}_{\mathcal{M}_j} = 1$ if $\mathcal{M}_j$ is the true model.

To achieve the desired amortization over any set of context variables $C_T$, we define a prior distribution $p(C_T)$ over the domains of $C_T$ and minimize the *context-aware* (CA) loss:

$$\mathcal{L}^{\text{CA}}(\boldsymbol{\phi}, \boldsymbol{\psi}) = \mathbb{E}\big[\mathcal{L}(\boldsymbol{\phi}, \boldsymbol{\psi}; C_T)\big], \tag{8}$$

where the (outer) expectation runs over $p(C_T)$ and $\mathcal{L}$ can be either $\mathcal{L}^{\text{PE}}$ or $\mathcal{L}^{\text{BMC}}$. We believe that uniform distributions are a reasonable choice of $p(C_T)$ for sensitivity analyses and employ them in all experiments. Nevertheless, $p(C_T)$ can be tailored to specific modeling needs, such as giving more weight to approximating a preferred baseline setting. In practice, we approximate Eq. 8 using standard mini-batch gradient descent over a finite data set $\mathcal{D} = \{C_T, \boldsymbol{\theta}, \boldsymbol{x}\}$ for parameter estimation, or $\mathcal{D} = \{C_T, \mathcal{M}_j, \boldsymbol{x}\}$ for model comparison.

Our approach seamlessly generalizes to other *strictly proper losses* (Gneiting & Raftery, 2007) which can be used as training objectives for amortized inference (Pacchiardi & Dutta, 2021). For the sake of generality, we can introduce a function $S$ that quantifies the fidelity of a conditional distribution $q_{\boldsymbol{\phi}} \in \mathcal{Q}$ for predicting a target quantity $\boldsymbol{y} \in \mathcal{Y}$

Table 1: Overview of our taxonomy for sensitivity in Bayesian inference via context variables. The rightmost column conveys that our context-aware (CA) loss function $\mathcal{L}^{\text{CA}}$ in Eq. 8 enables the amortization over both likelihood ($C_L$) and prior ($C_P$) contexts during training.

|  | Context | Sensitivity source example | Implementation | $\mathcal{L}^{\text{CA}}$ required? |
|---|---|---|---|---|
| $C_L$ | Likelihood | Structural model assumptions | Multiple simulator configurations | ✓ |
| $C_P$ | Prior | Expert knowledge | Multiple prior configurations | ✓ |
| $C_A$ | Approximator | Simulation gaps | Deep ensemble | ✗ |
| $C_D$ | Data | Influential observations | Multiple data configurations | ✗ |

(Gneiting & Raftery, 2007). Thus, $S : \mathcal{Q} \times \mathcal{Y} \to \mathbb{R}$ is a function of some $q_{\boldsymbol{\phi}}$ and $\boldsymbol{y}$ (e.g., $\boldsymbol{\theta}$ or $\mathcal{M}_j$) which can easily be written to incorporate arbitrary conditions for $q_{\boldsymbol{\phi}}$, such as summarized data $h_{\boldsymbol{\psi}}(\boldsymbol{x})$ and context $C_T$, resulting in $S(q_{\boldsymbol{\phi}}(\boldsymbol{y} \,|\, h_{\boldsymbol{\psi}}(\boldsymbol{x}), C_T), \boldsymbol{y})$. Ideally, we would like to treat the *expected score* as an optimization objective

$$\mathcal{L}(\boldsymbol{\phi}, \boldsymbol{\psi}; C_T) = \mathbb{E}_{p^*(\boldsymbol{x}, \boldsymbol{y})}\Big[S(q_{\boldsymbol{\phi}}(\boldsymbol{y} \,|\, h_{\boldsymbol{\psi}}(\boldsymbol{x}), C_T), \boldsymbol{y})\Big], \tag{9}$$

which for *strictly proper scoring functions* $S$ would guarantee $q_{\boldsymbol{\phi}}(\boldsymbol{y} \,|\, h_{\boldsymbol{\psi}}(\boldsymbol{x}), C_T) = p^*(\boldsymbol{y} \,|\, \boldsymbol{x})$ under perfect convergence (Gneiting & Raftery, 2007; Pacchiardi & Dutta, 2021). However, we usually cannot directly access the analytic expectation over the unknown true data-generating distribution $p^*(\boldsymbol{x}, \boldsymbol{y})$. Thus, to achieve tractable amortization, we use the model-implied distribution $p(\boldsymbol{x}, \boldsymbol{y} \,|\, \mathcal{M})$ as a proxy for the (unknown) true data generating process $p^*(\boldsymbol{x}, \boldsymbol{y})$ and optimize the former objective in expectation over the simulator outputs (i.e., one or more Bayesian probabilistic models).

With this expansion of the amortization scope, we achieve amortization across all sensitivity dimensions $C = (C_L, C_P, C_A, C_D)$ during inference. What that means in practice is that, at inference time, we can simply "turn a knob" on any of the sensitivity dimensions and obtain the resulting posterior in an instant. A natural question that immediately arises is *whether the resulting sensitivity-aware posterior is less accurate than the corresponding fixed-context posterior*. Intuitively, the answer depends on the sampling diversity of the contextualized joint model $p(\boldsymbol{x}, \boldsymbol{y}, C_T)$ and the potential for reaping the benefits of weight sharing: If the associated likelihood and prior variations instantiate generative models with wildly different behaviors, weight sharing may not be advantageous, resulting in diminishing returns from amortized training over the context variables $C_T$. Fortunately, the set of plausible choices for a given modeling problem typically leads to similar generative patterns, so that weight sharing is much more efficient than separate approximation. Indeed, our experiments demonstrate this for several representative model families even under small simulation budgets. Nevertheless, if amortization over very different simulators is desired, we recommend increasing the expressiveness of the neural approximator, the simulation budget, and the allotted training time. The following section describes sensitivity sources and actionable manipulation strategies for each sensitivity dimension (see Table 1 for an overview).

## 3.2 Sources of Sensitivity

**Likelihood and Prior Sensitivity**    Varying likelihood context variables are ubiquitous in simulation-based inference. Structural decisions within the simulator(s) may constitute context variables (e.g., the underlying scientific model, see **Experiment 2**) or exogenous experimental factors, such as design matrices, indicator variables, or time scales. Typical examples for prior context might be as simple as the scales of prior distributions, or as complex as different experts eliciting discrete sets of qualitative (e.g., non-probabilistic) domain knowledge. Moreover, both likelihood and prior can be continuously tempered to strengthen or weaken their influence. For example, power-scaling exponentiates densities with a parameter $\gamma > 0$, resulting in $p(\boldsymbol{\theta})^{\gamma}$ for prior power scaling and $p(\boldsymbol{y} \,|\, \boldsymbol{\theta})^{\gamma}$ for likelihood power scaling (Kallioinen et al., 2021).

We make all known pieces of likelihood and prior context explicit by incorporating $C_L$ and $C_P$ into the generative model. This enables amortization over these context variables, which drastically increases the generalization space of the trained neural approximator. During inference, the specific set of likelihood and prior can simply be selected by passing the respective $C_L$ and $C_P$ configurations. Amortization over the context space leverages structural similarities

between context configurations via weight sharing. Compared to separate training, this *minimizes the associated computational cost and is especially beneficial whenever only finite training data is available* (see **Experiment 2**).

**Approximator Sensitivity**  We define approximator sensitivity as the variability of inferential results due to the approximation method employed. To isolate approximator sensitivity in ABI, it seems helpful to (i) distinguish between a *closed world* (i.e., simulations) and *open world* (i.e., empirical data) setting; and (ii) realize an approximator context $C_A$ via a deep ensemble of $M$ equally configured and independently trained neural networks $\{(\phi^{(m)}), \psi^{(m)}\}_{m=1}^{M}$. [1]

In the closed-world setting, ground truth values for the approximation targets $y$ (i.e., $\theta$ or $\mathcal{M}_j$) are available. Thus, we can readily validate amortized neural approximators on thousands of simulated data sets from the model(s) under consideration. We propose to additionally measure performance variability between the ensemble members to detect approximator sensitivity due to finite training or suboptimal convergence. After validating the approximator in the closed world, we consider the open-world setting, where the true data-generating process is unknown.

As a simulation-based method, ABI assumes that simulations are a faithful representation of a system's real behavior (Dellaporta et al., 2022). Hence, *simulation gaps*, where atypical data violate this assumption, threaten its credibility (Schmitt et al., 2023; Cannon et al., 2022). Simulation gaps can be considered to cause an out-of-distribution (OOD) setting at inference time: For example, Cannon et al. (2022) observed that misspecification-induced simulation gaps result in neural approximators exhibiting the typical OOD behavior of unstable predictions (Ji et al., 2022; Shamir et al., 2021; Liu et al., 2021). Therefore, we can leverage the proven OOD detection capabilities of deep ensembles (Lakshminarayanan et al., 2017; Fort et al., 2019; Yang et al., 2022) to detect simulation gaps in ABI. Specifically, we hypothesize that *variability* across the $M$ ensemble members in the open world *despite consistent performance in the closed world* indicates a simulation gap. Concretely, we expect a simulation gap to translate into high variability in the predictive distribution of the unknown targets $y$ given empirical data $x_{\text{obs}}$,

$$\tilde{q}(y \mid x_{\text{obs}}) = \mathbb{E}_{p(\phi, \psi \mid \mathcal{D})} \left[ q_\phi(y \mid h_\psi(x_{\text{obs}})) \right] \tag{10}$$

which is approximated by the deep ensemble (Lakshminarayanan et al., 2017) and can be augmented with arbitrary context $C_T$.

When we train a deep ensemble with $M$ members, we clearly need to repeat the training loop $M$ times. Crucially though, we can simulate a single training set upfront and then re-use the simulated training data for all ensemble members. This not only reduces the stochastic dependence by keeping the training data constant but also drastically reduces the computational cost in most realistic tasks where simulations are expensive. Finally, our ensemble approach can be easily extended to combine information from all ensemble members for potentially more accurate inference (e.g., via simulation-based stacking, Yao et al., 2023) or investigate hyperparameter sensitivity via hyperparameter ensembles (Wenzel et al., 2020). Hyperparameters that are particularly relevant in SBI include the architecture of the summary network (e.g., inductive bias induced by the architecture, number of learned summary statistics), the choice of inference network (e.g., architecture, number of trainable weights), and common hyperparameters that are ubiquitous in deep learning (e.g., learning rate, regularization).

**Data Sensitivity**  Statistical inference relies on finite samples to draw conclusions about populations. As such, any analysis is influenced by the properties of this particular sample. For instance, analysis outcomes might radically change under different preprocessing choices, such as handling extreme or missing data points, even if these choices only affect a small subset (Simmons et al., 2011; Broderick et al., 2023).

There are two straightforward strategies to assess this data sensitivity: (i) To assess a disproportionally large influence of single data points (also known as influential observations), a context $C_D$ of alternative data manifestations can be realized via small perturbations of the empirical data set, for instance via bootstrapping or leave-one-out folds; (ii) to analyze the effect of specific preprocessing decisions, we can generate data set variations for all combinations of reasonable decisions, which in turn constitutes $C_D$. For example, Kristanto et al. (2024) identify 17 debatable preprocessing steps with 102 choice options in graph-based fMRI analysis, resulting in hundreds of potential manifestations of the final data set.

---

[1] Although Bayesian neural networks offer appealing uncertainty quantification properties, we focus on deep ensembles for their practical implementation advantages (Lakshminarayanan et al., 2017; Wilson & Izmailov, 2020).

Both strategies require a large amount of model refits, which is computationally infeasible for MCMC or non-amortized simulation-based approximators. In contrast, ABI methods can amortize across data sets of variable sizes (Radev et al., 2020), enabling rapid inference on a large number of data set variations.

### 3.3 Evaluating Sensitivity

**Quantitative Sensitivity**   We can easily *quantify* sensitivity via a divergence metric $\mathbb{D}$ between target probability densities (Kallioinen et al., 2021; Roos et al., 2015): For an acceptable upper bound $\vartheta$ based on domain knowledge, a model is robust if

$$\mathbb{D}\big[p(g(\boldsymbol{y}) \mid \boldsymbol{x}, C_i) \,\big|\big|\, p(g(\boldsymbol{y}) \mid \boldsymbol{x}, C_j))\big] < \vartheta, \tag{11}$$

for arbitrary context realizations $C_i$ and $C_j$, where $g(\boldsymbol{y})$ is a pushforward variable (e.g., predicted quantities) or a projection of the full target posterior onto a subset $\boldsymbol{y}' \subseteq \boldsymbol{y}$.

Measures from the family of $\mathcal{F}$-divergences offer principled metrics for $\mathbb{D}$ (Csiszár, 1964; Ali & Silvey, 1966; Liese & Vajda, 2006). In Bayesian model comparison, the model posterior containing the probabilities for each $\mathcal{M}_j$ follows a discrete categorical distribution. Thus, obtaining $\mathcal{F}$-divergences, such as the KL divergence, is straightforward (see **Experiment 3**, Figure 5b). In Bayesian parameter estimation, the posterior is typically not available as a closed-form density but as random draws. Thus, we prefer a probability integral metric, such as the maximum mean discrepancy (MMD; Gretton et al., 2012), which can be efficiently estimated from posterior samples (see Bischoff et al., 2024, for a recent discussion of other suitable choices).

**Qualitative Sensitivity**   Although quantitative sensitivity patterns provide detailed insights, sensitivity analysis is often ultimately interested in *qualitative* robustness, i.e., invariance of analytical conclusions to the context $C$ (Kallioinen et al., 2021; Bürkner et al., 2022). For instance, an analyst might ask whether two choices of context variables $C_1$ and $C_2$ contain a certain parameter value within a specified highest density interval (HDI), or lead to the selection of the same model $\mathcal{M}_j$. Making decisions based on a posterior distribution can be formalized via a decision function $L : \mathcal{P} \to \mathcal{A}$ which maps distributions $p \in \mathcal{P}$ (or their approximations) to possible qualitative conclusions or actions for a given problem $a \in \mathcal{A}$. Formally, qualitative robustness is expressed with the indicator function

$$R(C_i, C_j) = \begin{cases} 1 & \text{if } L\big(p(\boldsymbol{y} \mid \boldsymbol{x}, C_i)\big) = L\big(p(\boldsymbol{y} \mid \boldsymbol{x}, C_j)\big) \\ 0 & \text{otherwise} \end{cases} \tag{12}$$

which yields 1 if a conclusion is invariant to the choice of arbitrary context realizations $C_i$ or $C_j$, and 0 otherwise. Note that our definition trivially generalizes to more than two choices of context variables $C$.

## 4   Related Work

**Extending the amortization scope**   Wu et al. (2020) proposed a variational inference (VI) algorithm that amortizes over a family of probabilistic generative models. This meta-amortized VI approach learns transferable representations and generalizes to unseen distributions within the amortized family. The dependence on an analytically tractable likelihood function makes this approach inapplicable to simulation-based inference, while sequential approaches that enable likelihood-free VI (Wiqvist et al., 2021; Glöckler et al., 2022) lack the amortization properties essential for sensitivity analysis (see Table 2). Schröder & Macke (2023) perform amortized inference on a *set of models* by combining model comparison and parameter estimation into a single mixture generative model. On a related note, Dax et al. (2021) avoid training separate neural approximators for gravitational-wave parameter estimation by conditioning on detector-noise characteristics. Our SA-ABI method integrates ideas from amortization scope extension in a unified framework, enabling amortization over *any* plausible prior, likelihood, and data configurations while also assessing approximator sensitivity.

**Likelihood and prior sensitivity**   The posterior distribution clearly depends on the likelihood and prior, and a large body of research has studied the sensitivity to both likelihood and prior (for an overview, see Insua & Ruggeri, 2012; Depaoli et al., 2020). In non-amortized Bayesian inference, several approaches aim to avoid costly model refits by estimating the effects of local likelihood or prior perturbations on a given posterior, for example, via the infinitesimal jackknife (Giordano et al., 2018; 2019) or Pareto-smoothed importance sampling (Kallioinen et al., 2021). However,

Table 2: Comparison of the suitability of posterior approximation methods for sensitivity analysis in Bayesian inference.

| | VI | MAVI | SNVI | MCMC | IJ | IS | ABI | **SA-ABI** |
|---|---|---|---|---|---|---|---|---|
| Can handle intractable likelihoods | ✗ | ✗ | ✓ | ✗ | ✗ | ✗ | ✓ | ✓ |
| Amortized likelihood & prior sensitivity | ✗ | ✓ | ✗ | ✗ | ✓ | ✓ | ✗ | ✓ |
| Amortized data sensitivity | ✗ | ✓ | ✗ | ✗ | ✗ | ✗ | ✓ | ✓ |

VI: Variational Inference; MAVI: Meta-Amortized Variational Inference; SNVI: Sequential Neural Variational Inference; MCMC: Markov Chain Monte Carlo; IJ: Infinitesimal Jackknife; IS: Importance Sampling; ABI: Amortized Bayesian Inference; SA-ABI: Sensitivity-Aware ABI (ours).

these approaches require an analytically tractable likelihood function and rather expensive refits to evaluate data sensitivity. SA-ABI, in contrast, allows for a *direct* assessment of posteriors under different $C_L$ and $C_P$ contexts while eliminating likelihood tractability restrictions and the computational burden of refits (see Table 2). Our approach allows sensitivity analyses under drastic perturbations, which is not possible via established methods that rely on MCMC and importance sampling (e.g., when the scaling factor $\gamma$ approaches zero; Kallioinen et al., 2021). In **Experiment 3**, we demonstrate how our method enables prior sensitivity analyses up to a scaling factor of $\gamma = 0.1$.

**Approximator sensitivity**    Recent work has employed deep ensembles for posterior approximation (Balabanov et al., 2023; Tiulpin & Blaschko, 2022) or, within the scope of simulation-based inference, for improving estimation performance (Modi et al., 2023; Cannon et al., 2022), but not for quantifying the sensitivity induced by the approximation procedure. Schmitt et al. (2023) developed a method to detect simulation gaps in amortized Bayesian parameter estimation via out-of-distribution detection. We adopt a similar perspective on simulation gaps but focus on quantifying the resulting sensitivity in both parameter estimation and model comparison based on the variability of ensemble members. Beyond the identification of simulation gaps, SA-ABI has the crucial advantage of directly assessing the *real-life impact* of a simulation gap in terms of unreliable approximation.

**Data sensitivity**    The fact that analyses are sensitive to the input sample (i.e., data sensitivity) has wide-ranging implications across the sciences. First, the immoderate influence of single data points is closely related to traditional notions of robustness and simulation-based solutions thereof (Huang et al., 2023; Ward et al., 2022). Framed as a hostile scenario, adversarial attacks intend to exploit data sensitivity (Goodfellow et al., 2015; Biggio et al., 2012; Baruch et al., 2019), and adversarial robustness tries to prevent this (see Glöckler et al., 2023, for an ABI application). Second, the sensitivity to different preprocessing choices is directly linked to the reproducibility crisis in the empirical sciences (OSC, 2015; Wicherts et al., 2016). To render this sensitivity tangible, Steegen et al. (2016) introduced the *multiverse analysis*, which repeats an analysis across all alternatively processed data sets. The concept of a holistic analysis across plausible data configurations has been continually extended but is typically restricted by the computational feasibility of large-scale refits (Hall et al., 2022; Liu et al., 2020) or to specific estimation procedures (Broderick et al., 2023). In summary, our sensitivity-aware method unlocks (i) near-instant analyses of data sensitivity and adversarial susceptibility; and (ii) rapid multiverse analyses across a wide space of data processing decisions.

# 5   Experiments

In the following, we demonstrate the utility of our SA-ABI approach on applied, real-data modeling problems of COVID-19 outbreak dynamics (**Experiment 1**; prior sensitivity), climate modeling (**Experiment 2**; prior and likelihood sensitivity), and human decision-making (**Experiment 3**; prior, approximator, and data sensitivity). In each experiment, we first ensure the trustworthiness of SA-ABI by benchmarking it against standard ABI as the state-of-the-art approach for amortized inference on simulation-based models. Afterward, we use our validated approach to obtain insights into sensitivity-induced uncertainties that would have been hardly feasible with existing methods.

All implementations use the BayesFlow library for amortized Bayesian workflows (Radev et al., 2023b). Details for all experiments, such as model setup, network architecture and training, and additional results are available in the **Supplementary Material**.

Table 3: **Experiment 1**: Benchmarking approximation quality and time between standard ABI and SA-ABI (ours).

| Simulation budget | Method | MAE ↓ (± SD) | ECE ↓ (± SD) | PC ↑ (± SD) | Time by # of priors ↓ | | |
|---|---|---|---|---|---|---|---|
| | | | | | 1 | 3 | 1 000 |
| $2^{12} = 4\,096$ | ABI | $\mathbf{5.63} \pm 0.07$ | $0.009 \pm 0.0001$ | $0.27 \pm 0.05$ | **2min** | 6min | 1 876min |
| | **SA-ABI** | $5.69 \pm 0.06$ | $\mathbf{0.005} \pm 0.001$ | $\mathbf{0.28} \pm 0.01$ | **2min** | **2min** | **22min** |
| $2^{14} = 16\,384$ | ABI | $\mathbf{5.42} \pm 0.04$ | $0.008 \pm 0.001$ | $\mathbf{0.38} \pm 0.006$ | **6min** | 17min | 5 557min |
| | **SA-ABI** | $5.53 \pm 0.05$ | $\mathbf{0.006} \pm 0.002$ | $0.35 \pm 0.005$ | **6min** | **6min** | **26min** |
| $2^{16} = 65\,536$ | ABI | $\mathbf{5.37} \pm 0.005$ | $0.01 \pm 0.001$ | $\mathbf{0.40} \pm 0.007$ | **21**min | 62min | 20 721min |
| | **SA-ABI** | $5.44 \pm 0.006$ | $\mathbf{0.009} \pm 0.001$ | $0.39 \pm 0.01$ | **21min** | **21min** | **41min** |

*Note.* SD = Standard Deviation. MAE = Mean Absolute Error. ECE = Expected Calibration Error. PC = Posterior Contraction. Metrics are evaluated on the prior scaling setting $\gamma = 1.0$ with $N = 1\,000$ held-out data sets and averaged over ensembles of size $M = 2$ for each method. Thus, SDs reflect the within-ensemble variability. Total times for training and inference for $M = 1$ are reported (extrapolated for 1 000 prior sensitivity evaluations).

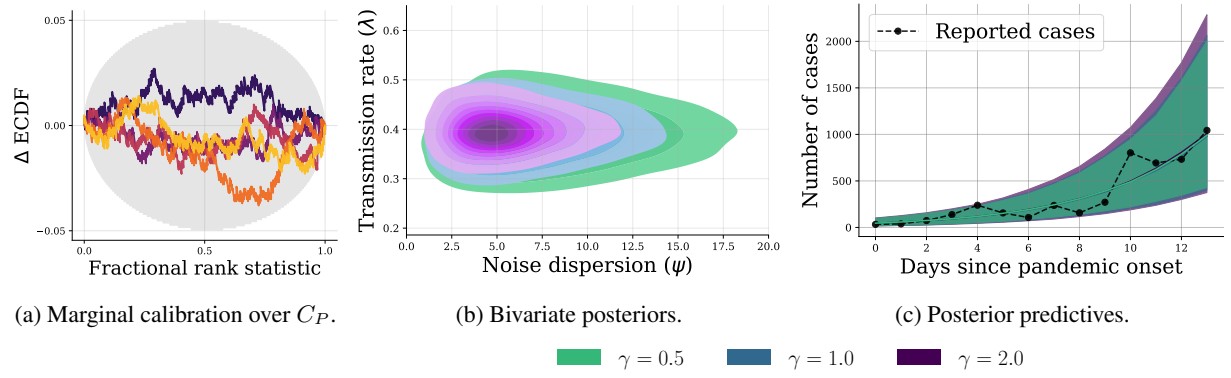

(a) Marginal calibration over $C_P$.     (b) Bivariate posteriors.     (c) Posterior predictives.

$\gamma = 0.5$     $\gamma = 1.0$     $\gamma = 2.0$

Figure 2: **Experiment 1.** (a) All parameters obtained by SA-ABI are well-calibrated over the full context space $C_P$. (b) The bivariate posterior of best recoverable parameters $\lambda$ and $\psi$ indicates substantial sensitivity in terms of uncertainty reduction for $\psi$, but (c) the posterior predictive distribution appears robust (overlapping median prediction lines and 90% CIs).

## 5.1 Experiment 1: COVID-19 Outbreak Dynamics

We set the stage with a straightforward example: Modeling the very early stage of a disease outbreak via a SIR model (Dehning et al., 2020). We demonstrate that obtaining amortized prior sensitivity insights does not compromise the approximation performance compared to standard ABI with the same fixed simulation budget. We adopt the simulation model from Radev et al. (2021b) and use a comparable neural network architecture specialized for time-series data. During training, we use power-scaling, that is, element-wise exponentiation $p(\boldsymbol{\theta})^\gamma$, to amortize over different priors. We sample the scaling factors in log space, $\boldsymbol{\gamma} \sim \exp(\mathcal{U}(\log(0.5), \log(2.0)))$, to ensure equal amounts of widening and shrinking. Thus, the prior context $C_P$ comprises a vector of scaling powers $\boldsymbol{\gamma}$ for each of the model parameters.

We first benchmark our SA-ABI approach against individual ABI instances trained solely on the tested baseline setting $\gamma = 1.0$. This allows us to determine the performance trade-off for the amortization scope expansion over $C_P$. Table 3 shows mostly comparable performance of our SA-ABI approach and individual ABI instances with only little trade-offs across all simulation budget settings, *despite SA-ABI spending only a fraction of the simulation budget on the tested settings*. The small variability within the deep ensembles further indicates approximator robustness in the simulated setting. Lastly, Table 3 highlights the time advantage of our method even for sensitivity analyses that only consider $C_P$.

Figure 2 shows the prior sensitivity results of our SA-ABI approach for the medium simulation budget of $N = 2^{14} = 16\,384$ simulations: Complementing the low calibration error in the benchmark setting without prior scaling, we also

Table 4: **Experiment 2**: Benchmarking approximation quality and time between standard ABI and SA-ABI (ours) in a limited data setting.

| Method | MAE ↓ (± SD) | ECE ↓ (± SD) | PC ↑ (± SD) | Time ↓ |
|---|---|---|---|---|
| ABI | $4.2 \pm 1.1$ | $0.08 \pm 0.07$ | $0.980 \pm 0.004$ | 313min |
| **SA-ABI** | $\mathbf{3.8 \pm 1.3}$ | $\mathbf{0.04 \pm 0.04}$ | $\mathbf{0.982 \pm 0.015}$ | **67min** |

*Note.* SD = Standard Deviation. MAE = Mean Absolute Error. ECE = Expected Calibration Error. PC = Posterior Contraction. Metrics are averaged over test data from all emission scenarios $\times$ climate model settings, resulting in 18 combinations with a total of $N = 2\,916$ held-out data sets. Thus, SDs reflect the variability of 18 individual results per method and metric. Total times for training and inference are reported. All networks use the uninformative prior context.

observe excellent calibration over the full prior space $C_P$ in Figure 2a. The bivariate posteriors for the two parameters with the best recovery (i.e., transmission rate and noise dispersion) unveil that prior sensitivity only affects the noise dispersion (see Figure 2b). Despite prior sensitivity in terms of parameter recoverability, model-based predictive performance is robust to prior scaling (see Figure 2c).

## 5.2 Experiment 2: Climate Trajectory Forecasting

In this experiment, we study whether model-based global warming forecasts are sensitive to the underlying climate model, emission scenario, and prior specification. Climate models estimate the solutions of differential equations for the fluid dynamics and thermodynamics of atmosphere, ocean, ice, and land masses. Since single forecasts can heavily depend on initial conditions, assumed emission scenarios, and the chosen climate model, modern global warming estimates build on a multitude of simulated trajectories (Riahi et al., 2017; Zelinka et al., 2020; Joshi et al., 2011).

However, trajectories simulated from climate models typically start in pre-industrial times, are not explicitly conditioned on any information since 1850, and are only available in a limited number. In their pioneering work, Diffenbaugh & Barnes (2023) combine neural networks trained on simulated trajectories with recent observational data to predict global warming trends and forecast when critical thresholds are reached. Here, we demonstrate the utility of our approach for *efficiently assessing the sensitivity of model-based predictions* in terms of qualitatively different assumptions regarding the underlying models, emission scenarios, and prior distributions.

Given a high-dimensional spatial observation dataset of surface temperatures (see Figure 3b, right), we are interested in temperature development predictions of different climate models under different future emission scenarios, specifically the time until a global mean surface temperature threshold is exceeded. Framed as a Bayesian parameter estimation task, we model the time-to-threshold $\theta$ and explicitly incorporate the climate model and an emission scenario (i.e., SSP1-3) in a likelihood context $C_L$. Furthermore, we encode a weakly informative prior $\theta \sim \mathcal{U}(-40, 41)$ that encompasses the full range of values present in the training data vs. an informative Gaussian prior $\mathcal{N}_+(10, 10)$, truncated to positive values based on the IPCC sixth assessment report (Lee et al., 2021), in a prior context $C_P$ (see Figure 3c. During training, the neural approximator learns to infer $\theta$ from simulated observations with the corresponding context. For each training example, we extract the ground truth $\theta$ from later stages of the simulated trajectory (see Figure 3b). At inference time, the network processes an unseen real observation $\boldsymbol{x}_{\text{obs}}$ from the year 2023 with a context that specifies an emission scenario, a climate model, and a prior configuration. The output is the contextualized approximate posterior $p(\theta \mid \boldsymbol{x}_{\text{obs}}, C)$.

We reproduce the results of Diffenbaugh & Barnes (2023) without sacrificing predictive accuracy (see **Supplementary Material**) and reveal sensitivity to the climate model. Further, Table 4 highlights the advantages of our joint training method utilizing information from *all* context configurations via weight sharing, which is especially relevant in the present limited data setting.

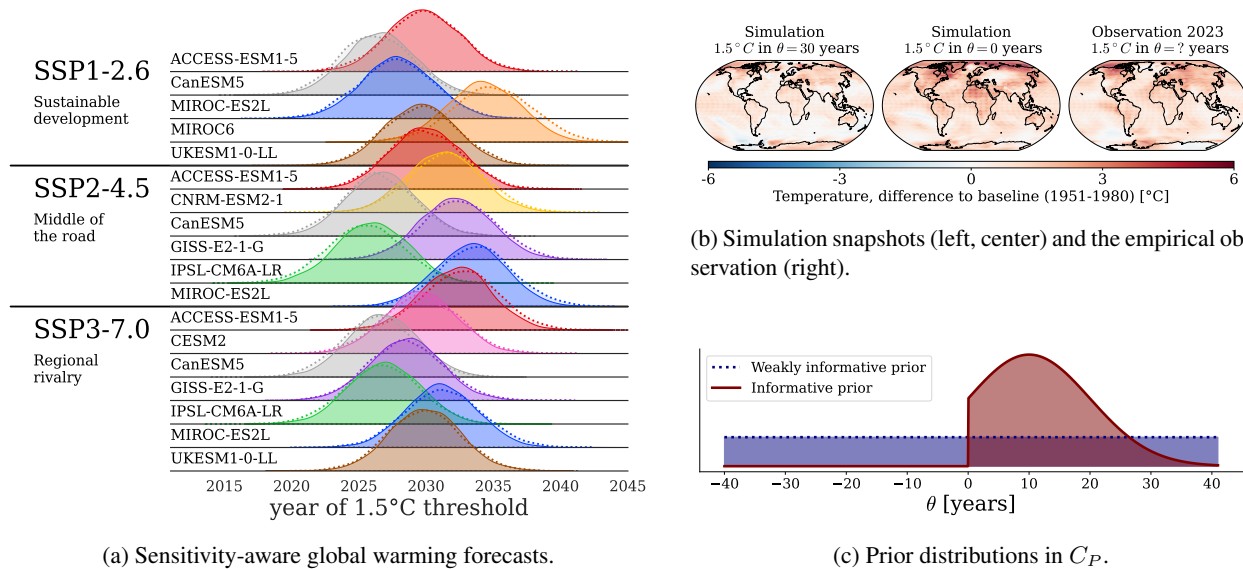

(a) Sensitivity-aware global warming forecasts.

(b) Simulation snapshots (left, center) and the empirical observation (right).

(c) Prior distributions in $C_P$.

Figure 3: **Experiment 2**. (a) Global warming forecasts are sensitive to the assumed climate model (rows) but not the emission scenario (SSP; groups of rows) or the prior (dotted: weakly informative prior; solid: informative prior). (b) Two examples of simulated observations from the climate model ACCESS-ESM1-5 (SSP-3) with known time-to-threshold (training data) and the current empirical observation that we use for the forecasts. (c) Prior predictive distributions of the weakly informative prior and the informative prior constituting the prior context $C_P$.

Finally, we study the sensitivity of climate forecasts for a real-world question of societal impact: *When will we reach the $1.5°C$ global warming threshold?* As depicted in Figure 3a, the answer to this question is sensitive to the underlying climate model but robust to the assumed emission scenario and informativeness of the prior. This finding is in accordance with Hawkins & Sutton (2009) who argue that the delayed effects of emission scenarios are unlikely to show on a short time scale such as the $1.5°C$ average warming threshold.

## 5.3 Experiment 3: Hierarchical Models of Decision-Making

This experiment extends our method to Bayesian model comparison of complex hierarchical models with analytically intractable likelihoods. The *drift-diffusion model* (DDM) is a popular stochastic model of decision-making widely used in cognitive science and neuroscience (Ratcliff et al., 2016). Elsemüller et al. (2023) recently compared four hierarchical models with ABI to test two proposed improvements of the DDM (referred to as $\mathcal{M}_1$): First, allowing model parameters to vary between experimental trials ($\mathcal{M}_3$ and $\mathcal{M}_4$) and second, allowing for evidence accumulation "jumps" via an additional parameter $\alpha$ ($\mathcal{M}_2$ and $\mathcal{M}_4$), which renders the likelihood function intractable.

Elsemüller et al. (2023) found clear evidence for inter-trial variabilities but unclear results concerning the utility of $\alpha$. In this experiment, we examine the sensitivity of these results to (i) the prior (via inference under 81 power-scaling perturbations of the hierarchical prior on $\alpha$), (ii) the approximator (via an ensemble of 20 equally configured neural networks), and (iii) the data (via 100 bootstrap samples). Thus, our comprehensive sensitivity analysis is based on $162\,000$ posterior model probabilities that are challenging to recover even *once* using existing methods. We now use the flexibility of our simulation-based approach to investigate the effects of shrinking and widening the hierarchical prior on $\alpha$ up to a factor of 10 and thus sample the $C_P$ scaling factors during training from $\gamma \sim \exp(\mathcal{U}(\log(0.1), \log(10.0)))$. The complexity of amortizing over the prior space is balanced by two aspects: While the scaling only affects the hyperpriors of the additional $\alpha$ parameter in $\mathcal{M}_2$ and $\mathcal{M}_4$, scaling up to a factor of 10 leads to much more extreme variations than the usual perturbations in likelihood-based settings of up to a factor of 2 (Kallioinen et al., 2021).

As before, we benchmark our SA-ABI approach against individual ABI instances trained solely on the tested baseline setting $\gamma = 1.0$, that is, without varying $C_P$ during training. Despite amortizing over a wide $C_P$ range, we observe little trade-offs in Table 5, with all ensemble members of both ABI and SA-ABI exhibiting near-perfect performance on

Table 5: **Experiment 3**: Benchmarking approximation quality and time between standard ABI and SA-ABI (ours) in a model comparison setting.

| Method | MAE ↓ (± SD) | ECE ↓ (± SD) | Accuracy ↑ (± SD) | Time by # of priors ↓ | |
|---|---|---|---|---|---|
| | | | | 1 | 1 000 |
| ABI | **0.012** ± 0.01 | **0.005** ± 0.002 | **0.99** ± 0.01 | **66min** | 66 349min |
| **SA-ABI** | 0.017 ± 0.01 | 0.01 ± 0.002 | 0.985 ± 0.01 | **66min** | **415min** |

*Note.* SD = Standard Deviation. MAE = Mean Absolute Error. ECE = Expected Calibration Error. Metrics are evaluated on the prior scaling setting $\gamma = 1.0$ with $N = 8\,000$ held-out data sets (2 000 per model) and averaged over ensembles of size $M = 20$ for each method. Thus, SDs reflect the within-ensemble variability. Total times for training and inference for $M = 1$ are reported (extrapolated for 1 000 prior sensitivity evaluations).

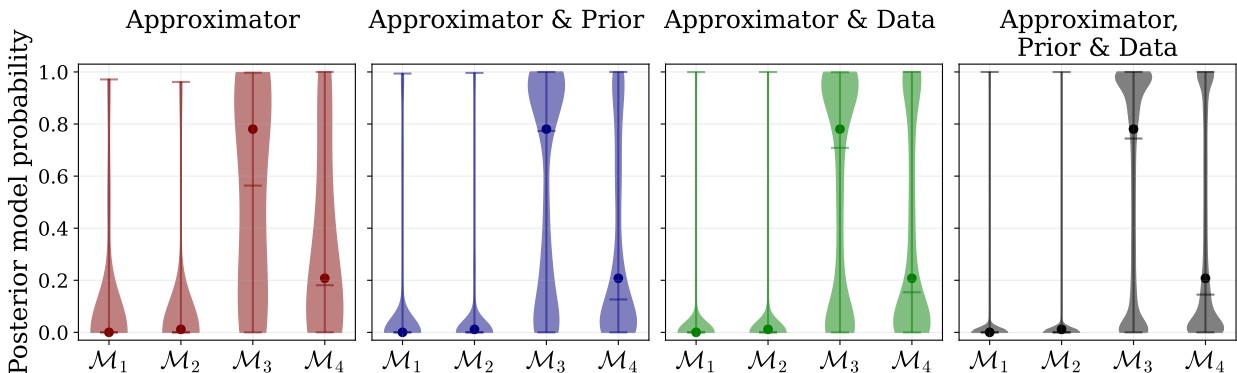

Figure 4: **Experiment 3**. Our sensitivity-aware posterior model probabilities indicate substantial approximator sensitivity but robustness to additional prior scaling and data perturbations. Dots represent the original results by Elsemüller et al. (2023).

simulated data. Strikingly, Figure 4 reveals highly inconsistent predictions of the ensemble members on the empirical data (also for ABI, see **Supplementary Material**). [2] This stark discrepancy implies the presence of a simulation gap. Indeed, contrasting the empirical data with the typical set of model simulations (Nalisnick et al., 2019; Morningstar et al., 2021) in Figure 5a flags the empirical data as out-of-distribution for the deep ensemble. Further, Figure 4 shows a comparatively low sensitivity against perturbing the hierarchical prior on $\alpha$ or the empirical data, with the medians under all perturbations close to the results by Elsemüller et al. (2023). Viewing deep ensemble predictions as approximate Bayesian model averaging (Wilson & Izmailov, 2020), we can conclude that the original results hold, but with substantial OOD uncertainty due to the simulation gap. A closer inspection of prior sensitivity in Figure 5b reveals (i) quantitative sensitivity to wide specifications of the hierarchical location $\mu_\alpha$, which increases under narrow specifications of the hierarchical scale $\sigma_\alpha$, and (ii) qualitative sensitivity to settings of $\mu_\alpha$ concerning the model with the highest probability, $\mathcal{M}_3$.

## 6 Conclusion

We proposed SA-ABI, an approach for large-scale sensitivity analyses with a keen emphasis on managing uncertainty in critical, high-impact scenarios. By leveraging amortized inference, our method causes minimal computational overhead during inference and can be directly integrated into software toolkits for amortized Bayesian workflows (such as Radev et al., 2023b; Tejero-Cantero et al., 2020). Future work should investigate more efficient approaches to quantify approximator sensitivity, with specific attention to Bayesian neural networks (Izmailov et al., 2021). Ad-

---

[2]Recall that the approximation targets in Bayesian model comparison are (categorical) posterior model probabilities, not posterior distributions over parameters. Thus, the variability shown in Figure 4 directly reflects the sensitivity caused by perturbing the respective model component(s).

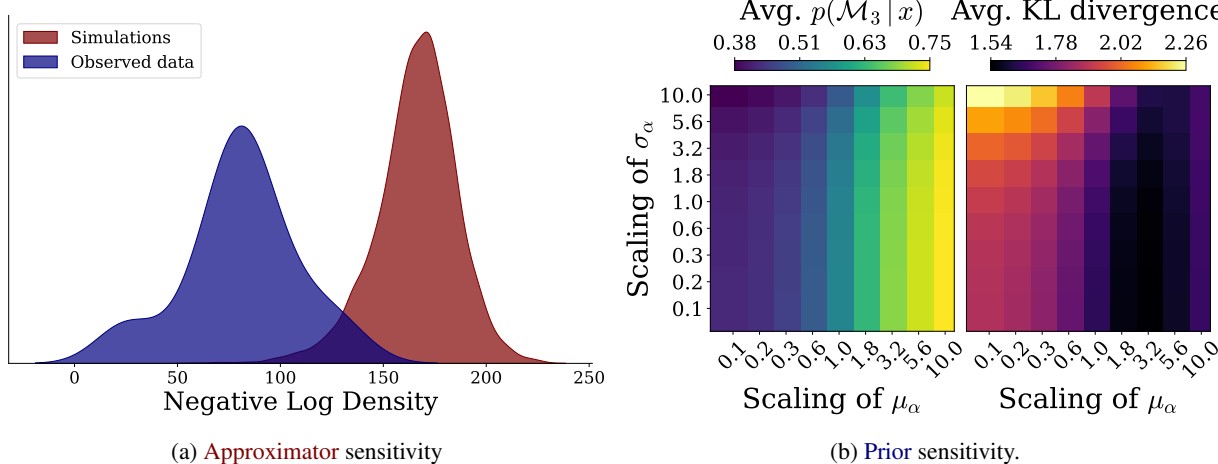

(a) Approximator sensitivity

(b) Prior sensitivity.

Figure 5: **Experiment 3**. (a) The learned summary statistics for the observed data are out-of-distribution (OOD) relative to the typical set of summary statistics for the model simulations (both distributions marginalized over $C_P$ and $C_A$). (b) The posteriors are quantitatively sensitive to power-scaling of the prior location $\mu_\alpha$, as indexed by the ensemble-averaged probability for $\mathcal{M}_3$ (left) as well as the KL divergence between the original results by Elsemüller et al. (2023) vs. scaled model posteriors (right). Notable qualitative sensitivity is present mainly due to different $\mu_\alpha$ values.

ditionally, whereas arbitrary data and approximator configurations can be explored at any time, likelihood and prior configurations have to be integrated into the training process. We believe that transfer learning (Bengio et al., 2009; Zhuang et al., 2021) is a promising tool to resolve this constraint, unlocking further flexibility on all sensitivity facets. We extend neural Bayesian inference (parameter estimation and model comparison) by amortizing over families of probabilistic models, as characterized by context variables $C$. This drastically expands the amortization scope of the employed neural approximators and constitutes a major leap towards foundation models for probabilistic (Bayesian) inference. Follow-up research in this direction might further increase the probabilistic model space during the training stage to facilitate near-universal amortized inference with pre-trained neural networks.

## Acknowledgments

We thank the anonymous reviewers and the action editor for improving the manuscript with their constructive and thoughtful feedback. Additionally, we thank Noah S. Diffenbaugh and Elizabeth A. Barnes for helping us build upon their pioneering work in Experiment 2. LE was supported by a grant from the Deutsche Forschungsgemeinschaft (DFG, German Research Foundation; GRK 2277) to the research training group Statistical Modeling in Psychology (SMiP) and the Google Cloud Research Credits program with the award GCP19980904. HO was supported by the state of Baden-Württemberg through bwHPC. MS was supported by the Cyber Valley Research Fund (grant number: CyVy-RF-2021-16) and the DFG under Germany's Excellence Strategy EXC-2075 - 390740016 (the Stuttgart Cluster of Excellence SimTech). UK was supported by the Informatics for Life initiative funded by the Klaus Tschira Foundation and the DFG under Germany's Excellence Strategy EXC-2181 - 390900948 (the Heidelberg Cluster of Excellence STRUCTURES).

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

# Supplementary Material

## A  Frequently Asked Questions (FAQ)

**Q: How can I reproduce the results?**

Code for reproducing all results from this paper is freely available at `https://github.com/bayesflow-org/SA-ABI`.

**Q: Can I apply your sensitivity-aware approach to posterior predictive model comparison as well?**

Yes! In this work, we focus on prior predictive model comparison, but our ideas directly apply to posterior predictive metrics (Gelman et al., 2014), such as leave-one-out cross-validation (Vehtari et al., 2017). We recommend the joint usage of a posterior and a likelihood network as proposed in Radev et al. (2023a) to achieve amortization in this application.

**Q: Are there limits to the distributional shapes that can be explored in $C_L$ and $C_P$?**

The only requirement for distributions in the likelihood and prior context is being able to simulate data from the resulting model. Besides that, SA-ABI gives modelers full flexibility in specifying any theoretically meaningful alternative formulations without concerns about analytically tractable likelihoods, unlike MCMC methods.

**Q: How exactly did you encode the context variables $C_L$ and $C_P$ in a suitable format for the neural network in your experiments?**

In **Experiment 1** and **Experiment 3**, each prior distribution over a parameter is continuously tempered by power-scaling. Consequently, $C_P$ is encoded by a vector holding the scaling factors for each prior component. In **Experiment 2**, both $C_L$ and $C_P$ consist of discrete choices and are therefore passed to the inference network in one-hot-encoded vectors.

## B  Methods: Additional Details

### B.1  Hypothesis Testing for Quantitative Sensitivity

We can employ a sampling-based (frequentist) hypothesis test to determine the probability of observed $\mathbb{D}(\cdot \,||\, \cdot)$ estimates (hitherto referred to as $\widehat{\mathbb{D}}$) under the null hypothesis of zero difference between $p(\boldsymbol{\theta} \,|\, \boldsymbol{x}, C_i)$ and $p(\boldsymbol{\theta} \,|\, \boldsymbol{x}, C_j)$. For this, we can construct an approximate sampling distribution of $\widehat{\mathbb{D}}$ under the null hypothesis via bootstrap or permutation tests based on multiple draws from $p(\boldsymbol{\theta} \,|\, \boldsymbol{x}, C_i)$. Based on the approximate sampling distribution, we can then obtain a critical $\widehat{\mathbb{D}}$ value for a fixed Type I error probability $\delta$ and compare it to the observed one. The power of such a test will generally be high when having access to many draws from $p(\boldsymbol{\theta} \,|\, \boldsymbol{x}, C_i)$ and $p(\boldsymbol{\theta} \,|\, \boldsymbol{x}, C_j)$, a requirement that is easily met in the context of ABI.

## C  Experiments: Implementation Details and Additional Results

### C.1  Benchmarking Metrics

For the benchmarks conducted in **Experiment 1** and **Experiment 2**, we measure three complementary performance metrics on $J$ unseen test data sets $\{\mathcal{D}_o^{(j)}\}_{j=1}^J$ with known ground-truth parameters $\{\boldsymbol{\theta}_*^{(j)}\}_{j=1}^J$. For each data set $\mathcal{D}_o^{(j)}$, we obtain a set $\{\boldsymbol{\theta}_s^{(j)}\}_{s=1}^S$ of $S$ posterior draws from the neural approximator $q_\phi(\boldsymbol{\theta} \,|\, \mathcal{D}_o^{(j)})$. We summarize each metric into a single measure across all test data sets and parameters for a given neural approximator and test setting (e.g., $\boldsymbol{\gamma} = 0.5$ in **Experiment 1**).

We use the *Mean Absolute Error* (MAE) to measure the overall error between posterior draws $\boldsymbol{\theta}_s^{(j)}$ and ground-truth parameters $\boldsymbol{\theta}_*^{(j)}$:

$$\text{MAE} = \frac{1}{J}\sum_{j=1}^{J}\left|\frac{1}{S}\sum_{s=1}^{S}\left(\boldsymbol{\theta}_s^{(j)} - \boldsymbol{\theta}_*^{(j)}\right)\right|. \tag{13}$$

Further, we assess uncertainty calibration via the *Expected Calibration Error* (ECE). In Bayesian parameter estimation, all uncertainty regions $U_q(\boldsymbol{\theta}\mid\mathcal{D})$ of the true posterior $p(\boldsymbol{\theta}\mid\mathcal{D})$ are by definition well-calibrated for any quantile $q \in (0,1)$ (Bürkner et al., 2022), such that:

$$q = \iint \mathbf{I}[\boldsymbol{\theta}_* \in U_q(\boldsymbol{\theta}\mid\mathcal{D})]\, p(\mathcal{D}\mid\boldsymbol{\theta}_*)\, p(\boldsymbol{\theta}_*)\mathrm{d}\boldsymbol{\theta}_*\mathrm{d}\mathcal{D}, \tag{14}$$

with $\mathbf{I}[\cdot]$ denoting the indicator function. Simulation-based calibration (SBC; Talts et al., 2018) measures miscalibration via deviations from this equality. We estimate the ECE via the median SBC error of 20 linearly spaced credible intervals with quantiles of $q \in [0.5\%, 99.5\%]$. [3] Lastly, we measure Bayesian information gain via the median *Posterior Contraction* (PC) across data sets, defined as $1 - \text{Var}(\text{Posterior})/\text{Var}(\text{Prior})$ (Betancourt, 2018).

## C.2 Experiment 1: COVID-19 Outbreak Dynamics

**Model Setup:** We consider a simple SIR model where individuals are either susceptible, $S$, infected, $I$, or recovered, $R$. Both infection and recovery are modeled with a constant transmission rate $\lambda$ and recovery rate $\mu$, respectively. The model is described by a system of ordinary differential equations (ODEs),

$$\frac{dS}{dt} = -\lambda\left(\frac{S\,I}{N}\right), \tag{15}$$

$$\frac{dI}{dt} = \lambda\left(\frac{S\,I}{N}\right) - \mu\,I, \tag{16}$$

$$\frac{dR}{dt} = \mu\,I, \tag{17}$$

with $N = S + I + R$ denoting the total population size. In addition to the ODE parameters $\lambda$ and $\mu$, our model includes a reporting delay parameter $D$ and a noise dispersion parameter $\psi$, which jointly influence the (noisy) number of reported infected individuals via

$$I_t^{(obs)} \sim \text{NegBinomial}(I_{t-D}^{(new)}, \psi), \tag{18}$$

with $I^{(new)} = \lambda(S_t I_t/N)$. The negative binomial distribution allows for modeling dispersion, i.e., variation of the variability independent of the mean, which is considered likely for early phases of the COVID-19 pandemic (Blumberg et al., 2014; Braumann et al., 2021). For our implementation, we transform the parameterization in Equation 18 with mean $I_{t-D}^{(new)}$ and dispersion $\psi$ to the `numpy` library's implementation of the negative binomial distribution with number of successes $n$ and probability of success $p$:

$$n = \psi \tag{19}$$

$$\sigma = \frac{I_{t-D}^{(new)} + 1}{\psi(I_{t-D}^{(new)})^2} \tag{20}$$

$$p = \frac{\sigma - I_{t-D}^{(new)}}{\sigma}. \tag{21}$$

The fifth estimated model parameter is the initial number of infected individuals $I_0$.

We use the same prior specification as Radev et al. (2021b), which is displayed in Table 6 along with the respective power-scaling scheme. Figure 6 shows the behavior of the prior predictive distributions under different power-scaling values $\gamma$.

---

[3] In **Experiment 3**, we use the ECE formulation by Naeini et al. (2015) for probabilistic classification.

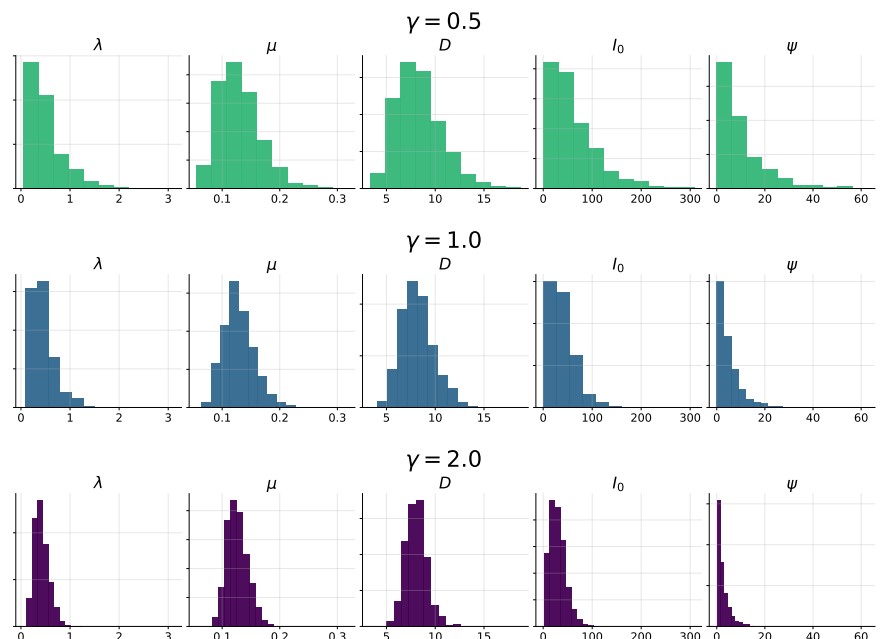

Figure 6: **Experiment 1**. Prior predictive distributions under different scaling parameters $\gamma$.

Table 6: **Experiment 1**. Power-scaled prior distributions for all parameters.

| Parameter | Symbol | Power-scaled prior distribution |
|---|---|---|
| Transmission rate | $\lambda$ | $\text{LogNormal}(\log(0.4), 0.5/\sqrt{\gamma_1})$ |
| Recovery rate of infected individuals | $\mu$ | $\text{LogNormal}(\log(1/8), 0.2/\sqrt{\gamma_2})$ |
| Reporting delay (lag) | $D$ | $\text{LogNormal}(\log(8), 0.2/\sqrt{\gamma_3})$ |
| Initial number of infected individuals | $I_0$ | $\text{Gamma}(2\gamma_4 - \gamma_4 + 1, 20/\gamma_4)$ |
| Noise dispersion | $\psi$ | $\text{Exponential}(5/\gamma_5)$ |

*Note*. Our parameterization follows the `numpy` library's implementation of the respective distribution.

We use time-series data from the first two weeks of the COVID-19 pandemic in Germany provided by the Center for Systems Science and Engineering (CSSE) at Johns Hopkins University, licensed under CC BY 4.0. [4]

**Neural Network and Training:** Our neural network architecture follows a simplified version of the design implemented by Radev et al. (2021b): We use a recurrent network with gated recurrent units as summary network and a conditional invertible network as inference network.

All computations for this experiment were performed on a single-GPU machine with an NVIDIA RTX 3070 graphics card and an AMD Ryzen 5 5600X processor. Simulating $16\,384$ training data sets took $8$ seconds and subsequent offline training for $75$ epochs took $6$ minutes.

**Additional Results:** We further investigate SA-ABI and ABI for potential reductions in approximation performance due to amortized prior sensitivity in the medium simulation budget setting of $2^{14} = 16\,384$. Figure 7 and Figure 8 show similar parameter recovery in the baseline $\gamma = 1.0$ setting, both limited by the small number of $T = 14$ data points available. Figure 9 and Figure 10 demonstrate that calibrated predictions are nevertheless mostly achievable, with equal patterns between SA-ABI (Figure 9) and standard ABI (Figure 10) for the baseline setting. Figure 11 additionally contains MMD hypothesis tests that clearly show sensitivity of the parameter posterior to the prior specification.

---

[4]https://raw.githubusercontent.com/CSSEGISandData/COVID-19/master/csse_covid_19_data/csse_covid_19_time_series/time_series_covid19_confirmed_global.csv

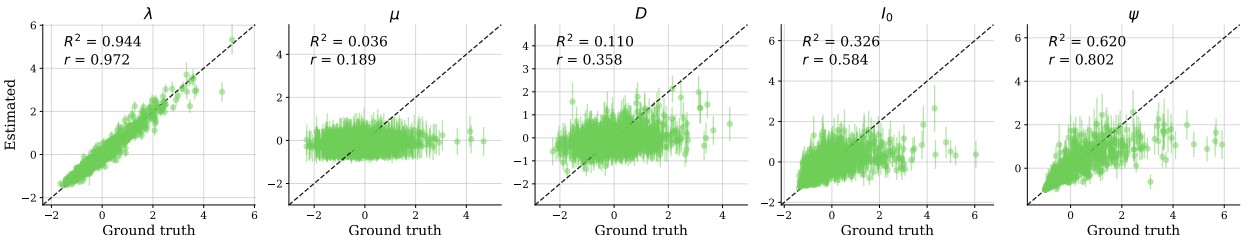

Figure 7: **Experiment 1**. Simulation-based recovery of the context-aware neural approximator used in our experiment for $\gamma = 1.0$ (simulation budget of $2^{14} = 16\,384$).

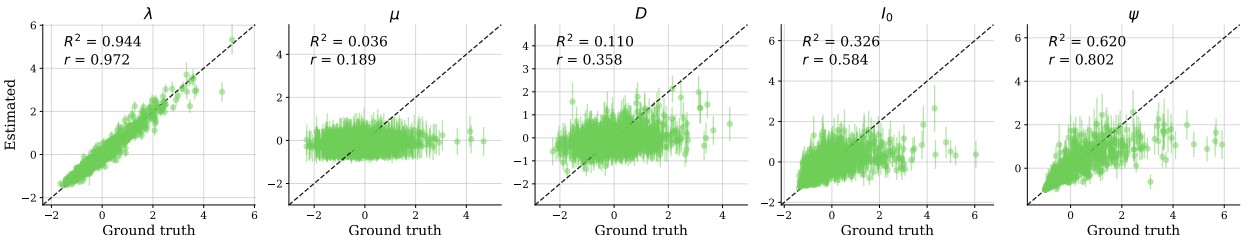

Figure 8: **Experiment 1**. Simulation-based parameter recovery of a single-prior neural approximator trained with the same configuration and simulation budget ($2^{14} = 16\,384$) as in our experiment but without $C_P$ (i.e., on the baseline $\gamma = 1.0$ setting).

**Benchmark Details:** All results of Experiment 1 except the benchmark operate in a standardized parameter space to align the different parameter scales. To eliminate the influence of standardization mismatches across the tested settings, the networks trained for the benchmark use the original (unstandardized) parameters. We further employ ensembles of size $M = 2$ to check for potential approximator sensitivity affecting the stability of the benchmarking results and observe stable results within the ensemble members (i.e., no approximator sensitivity). All networks are trained for 75 epochs.

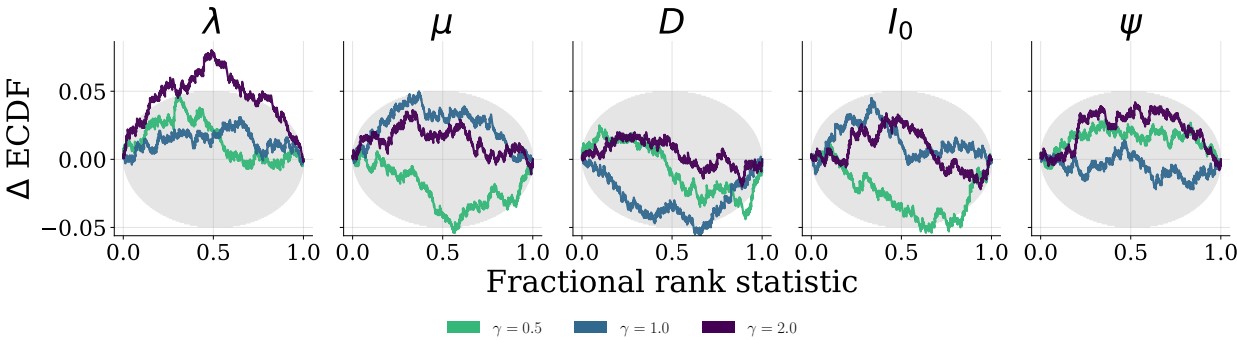

Figure 9: **Experiment 1**. Simulation-based calibration of the context-aware neural approximator used in our experiment for contexts $\gamma \in \{0.5, 1.0, 2.0\}$ (simulation budget of $2^{14} = 16\,384$).

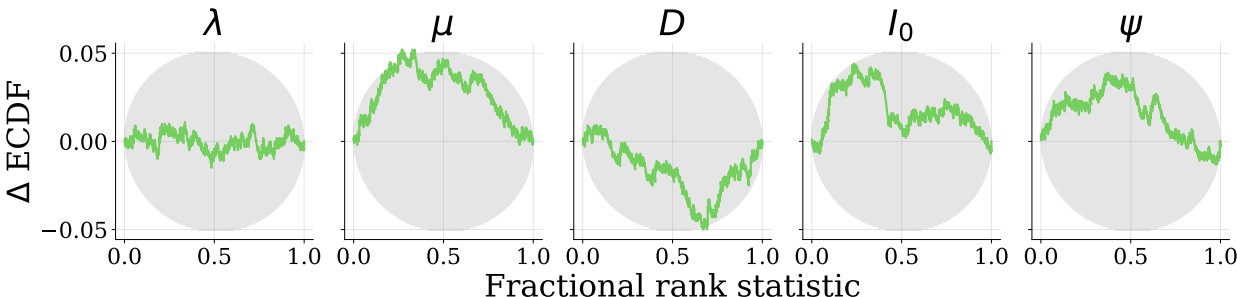

Figure 10: **Experiment 1**. Simulation-based calibration of a single-prior neural approximator trained with the same configuration and simulation budget ($2^{14} = 16\,384$) as in our experiment but without $C_P$ (i.e., on the baseline $\gamma = 1.0$ setting).

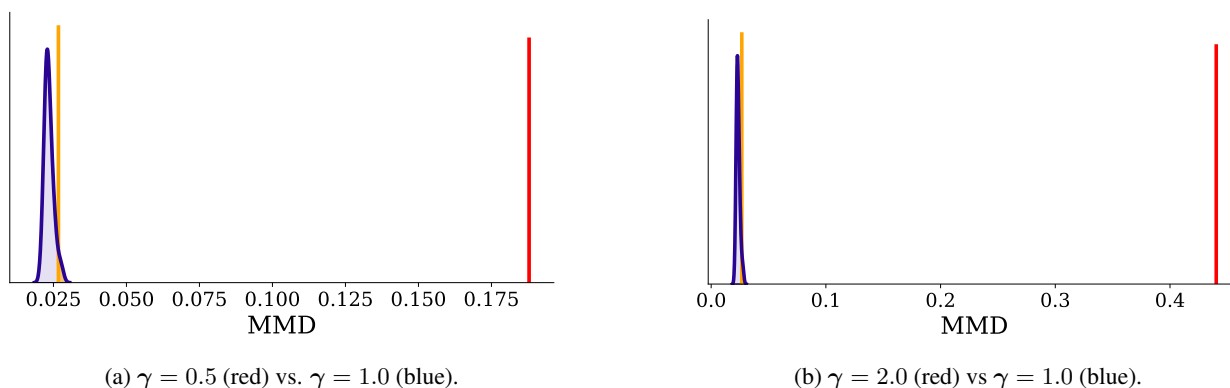

(a) $\gamma = 0.5$ (red) vs. $\gamma = 1.0$ (blue).

(b) $\gamma = 2.0$ (red) vs $\gamma = 1.0$ (blue).

Figure 11: **Experiment 1**. MMD hypothesis tests confirm sensitivity to prior specification in the parameter space. The blue density represents samples under the null distribution of zero difference between $\gamma = 1.0$ and the scaled posteriors (red; $\gamma = 0.5$ or $\gamma = 2.0$.), the yellow lines mark the area with a $\delta = 5\%$ rejection probability.

## C.3 Experiment 2: Climate Trajectory Forecasting

**Data:** Climate models (CM) are typically differential equations describing all relevant components of the Earth system and their couplings. Socioeconomic pathways (SSPs) provide comprehensive scenarios of future developments by describing qualitatively different trajectories of future emissions. Following standard practice, we append the associated radiative forcing in the year 2100 to the SSP identifier (e.g., SSP1-2.6 refers to the SSP1 scenario with radiative forcing of $2.6 W/m^2$). Given a CM, an SSP, and an initial condition (IC), high-dimensional trajectories of many observables can be simulated from the underlying model. Here we restrict the analysis to the air temperature at the surface (TAS).

Simulations produce trajectories $\text{TAS}^{\text{CM, SSP, IC}}(\vec{x}, t)$ where $\vec{x}$ is a spatial coordinate on the Earth's surface and $t$ is a time between 1850 and 2100. From this, the area-weighted global mean surface air temperature (GSAT) can be computed, which is the key indicator of global warming over pre-industrial levels.

All data used in this experiment is freely available to download: For the climate model simulation outputs, we use data from the Earth System Grid Federation.[5] For the observational data set for 2022, we use data from Berkeley Earth, licensed under CC BY 4.0.[6] We include all climate models that have archived at least 10 trajectories for the future emission scenarios SSP1, SSP2, and SSP3 in our analysis. We reshape all data to a $2.5 \times 2.5$ longitude-latitude grid and compute yearly differences to the baseline period (1951 to 1980). The warming threshold is defined relative to the pre-industrial period 1850 to 1900, from which we can directly obtain the time-to-threshold parameter $\theta$ for each tuple of year, model, and trajectory ensemble.

**Model Setup:** Our model setup focuses on the time $\theta$ until the $1.5°C$ warming threshold is reached, but can easily be adapted to arbitrary temperature thresholds. We realize a prior context $C_P$ for $\theta$ by two discrete prior choices: First, a weakly informative prior, $\theta \sim \mathcal{U}(-40, 41)$, that encompasses the full range of values present in the training data of simulated climate warming trajectories. Second, a more informative Gaussian prior, $\mathcal{N}_+(10, 10)$, truncated to include only positive values. This prior is based on the IPCC sixth assessment report stating that the central estimate of crossing the $1.5°C$ threshold lies in the early 2030s (Lee et al., 2021).

Table 7: Overview of the climate models included in each SSP emission scenario.

| Climate Models | SSP1-2.6 | SSP2-4.5 | SSP3-7.0 |
|---|---|---|---|
| ACCESS-ESM1-5 | ✓ | ✓ | ✓ |
| CanESM5 | ✓ | ✓ | ✓ |
| CESM2 | | | ✓ |
| CNRM-ESM2-1 | | ✓ | |
| GISS-E2-1-G | | ✓ | ✓ |
| IPSL-CM6A-LR | | ✓ | ✓ |
| MIROC-ES2L | ✓ | ✓ | ✓ |
| MIROC6 | ✓ | | |
| UKESM1-0-LL | ✓ | | ✓ |

The likelihood is obtained from the simulated trajectories of the climate models. For a given time-to-threshold $\theta$ and climate model (encoded in the likelihood context $C_L$), trajectories of the respective climate model are selected in ensembles of 10. We first identify the year of threshold exceedance of the mean global surface temperature across a trajectory ensemble. [7] Afterwards, a random ensemble and trajectory are chosen. Finally, the likelihood algorithm returns the spatial temperature pattern that is $\theta$ years prior to the year of threshold exceedance in the simulated trajectory.

---

[5] https://esgf.llnl.gov/

[6] https://berkeleyearth.org/data/

[7] Averaging over trajectory ensembles smoothes out the chaotic internal variability to obtain a more stable estimate. In contrast, the IPCC defines the year of threshold exceedance as the middle of a 20-year averaging window. Diffenbaugh & Barnes (2023) show that resulting forecasts are insensitive to the chosen definition.

**Neural Network and Training:**  We z-standardize data and parameters before passing them to the neural approximator. As summary network, we use a dense network that parallels the architecture used in Diffenbaugh & Barnes (2023): Inputs come in the form of 72x144 temperature grids and are flattened. Two hidden layers of 25 units with ReLU activation are followed by 8 output units of learned summary statistics. During training, we additionally employ dropout regularization with a dropout probability of $0.4$ on the initial layer of the summary network to mitigate overfitting. As inference network, we use a conditional invertible neural network. Since normalizing flows require more than one dimension, we add a dummy standard normal parameter.

Training the neural approximator took approximately 70 minutes on a consumer notebook with a 6-core AMD Ryzen 5 5625U CPU and without a dedicated graphics card, underscoring the wide applicability of our method.

For the individual ABI instances trained for the benchmark, each instance was trained on data of a specific scenario $\times$ climate model setting. We used the same network architecture for separate and joint training to enhance comparability, but note that the hyperparameters may not be optimal for the respective data size settings. Joint training was conducted on $80$ epochs, whereas we chose a smaller number of $15$ epochs for the separate training to mitigate overfitting.

**Additional Results:**  To validate our approach, we compute the mean absolute error (MAE) of the point estimate (posterior mean) $\hat{\theta}$ to the true value $\theta^*$ on a held-out validation set. Figure 12a shows good recovery across climate models, true time-to-threshold $\theta^*$, and SSPs. To parallel the evaluation procedure of Diffenbaugh & Barnes (2023), who did not differentiate between the climate models, we use a flat prior via the prior context $C_P$ and marginalize predictions over all climate models contained in $C_L$. Figure 12b shows that our sensitivity-aware approach does not sacrifice predictive accuracy and further enables the identification of biased estimates for MIROC6 in the SSP1-2.6 scenario as the main limitation to better performance. Figure 13 provides an additional perspective via standard simulation-based recovery and calibration plots. Overall recovery is high, but calibration plots indicate notable underconfidence of the posteriors, implying that the networks systematically overestimate the variance of time-to-threshold $\theta$ predictions. We hypothesize that this is due to an underperforming summary network which we nevertheless keep the same as in Diffenbaugh & Barnes (2023) for the sake of comparability.

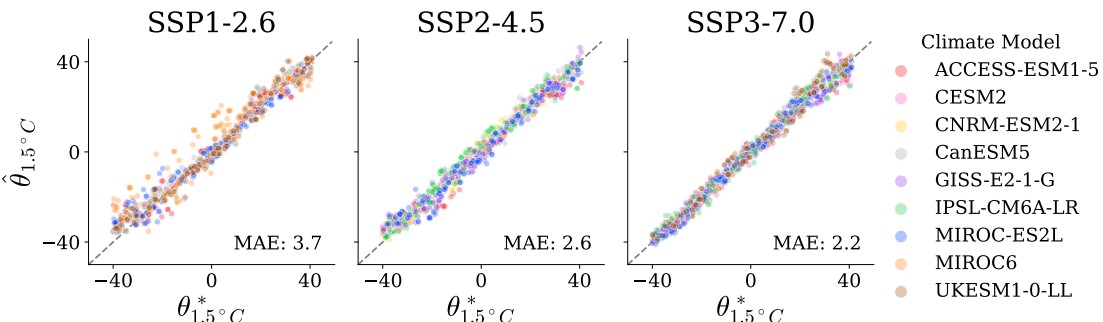

(a) Weakly informative prior. Recovery with $C_L$ information about the respective climate model given is good (total MAE of 2.0) across climate models, true time-to-threshold $\theta^*$, and SSPs.

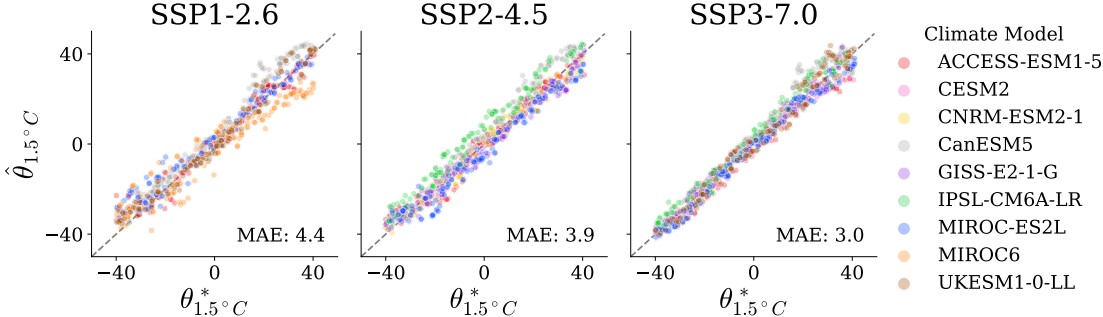

(b) Weakly informative prior. Recovery without $C_L$ information about the respective climate model given – all predictions are not only obtained for the appropriate climate model, but all climate models contained in $C_L$ and afterwards averaged. This leads to a validation setup comparable to Diffenbaugh & Barnes (2023). The mean absolute error (MAE) of 3.0 years for SSP3-7.0 indicates that our approach does not sacrifice predictive accuracy in comparison to Diffenbaugh & Barnes (2023), who report MAE between 2.7 and 3.8 years for the same task (eyeballed values from boxplots reported in the appendix).

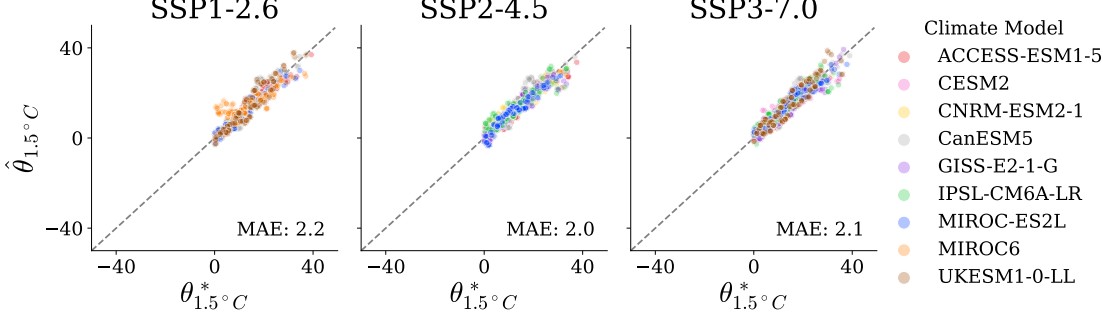

(c) Informative prior. Recovery with $C_L$ information about the respective climate model given, here restricted to positive $\theta$ values due to the truncation of the prior. Recovery is good with a total MAE of 1.2.

Figure 12: **Experiment 2**. Recovery of time-to-threshold on held-out validation data.

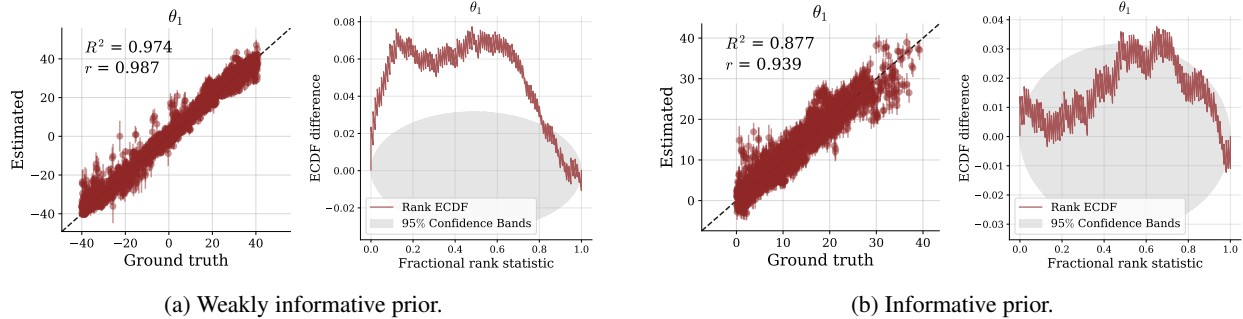

(a) Weakly informative prior.              (b) Informative prior.

Figure 13: **Experiment 2**. Standard simulation-based metrics of recovery and calibration on held-out validation data for both prior contexts $C_P$. All results are marginalized over the likelihood context space $C_L$ (i.e., climate models and emission scenarios).

### C.4 Experiment 3: Comparing Hierarchical Models of Decision-Making

**Model Setup:** The drift-diffusion model (DDM; Ratcliff et al., 2016; Ratcliff, 1978) models binary decision outcomes and their associated response times with the following stochastic ordinary differential equation:

$$\mathrm{d}x = v\,\mathrm{d}t + \xi\sqrt{\mathrm{d}t} \tag{22}$$

$$\xi \sim \mathcal{N}(0,1), \tag{23}$$

with $\mathrm{d}x$ denoting the change in evidence accumulation, $v$ denoting the rate of evidence accumulation, and $\xi$ the noise of evidence accumulation. Additional parameters of the model include the non-decision time $t_0$ (e.g., encoding and motor response), the decision threshold $a$, and the bias towards a decision option $z_r$.

The Lévy flight model (LFM; Voss et al., 2019) relaxes the Gaussian noise assumption by using the more general $\alpha$-stable distribution, leading to an unknown analytical form of the likelihood:

$$\mathrm{d}x = v\,\mathrm{d}t + \sigma\mathrm{d}\xi \tag{24}$$

$$\xi \sim \mathrm{AlphaStable}(\alpha, \mu = 0, \sigma = \frac{1}{\sqrt{2}}, \beta = 0), \tag{25}$$

with the additional stability parameter $\alpha$ which shall also be estimated.

Additionally, there is a debate about whether the model parameters $v$, $z_r$, and $t_o$ should have a fixed value over the course of an experiment (basic models) or be allowed to vary (full models) throughout the experiment (Lerche & Voss, 2016; Boehm et al., 2018). Therefore, we compare the following four models in this experiment:

- Basic DDM ($\mathcal{M}_1$): The classic four-parameter formulation with the model parameters $v$, $t_0$, $a$, and $z_r$.

- Basic LFM ($\mathcal{M}_2$): Equals the basic DDM plus the additional stability parameter $\alpha$ controlling the tail behavior of the noise distribution.

- Full DDM ($\mathcal{M}_3$): Equals the basic DDM plus inter-trial variability parameters $s_v$, $s_{t_0}$, and $s_{z_r}$.

- Full LFM ($\mathcal{M}_4$): Equals the full DDM plus the additional stability parameter $\alpha$.

As in Elsemüller et al. (2023), we reanalyze data by Wieschen et al. (2020) (provided by the original authors) containing 40 participants with 900 decision trials each and assume a uniform model prior (i.e., equal prior model probabilities). We use the same hierarchical priors as Elsemüller et al. (2023), who provided a detailed table with all prior choices. Since they are central to our experiment, we reiterate the priors leading to $\alpha_m$ for each participant $m$ here:

$$\begin{aligned}
\mu_\alpha &\sim \mathcal{N}(1.65, 0.15/\gamma_1) \\
\sigma_\alpha &\sim \mathcal{N}_+(0.3, 0.1/\gamma_2) \\
\alpha_m &\sim \mathcal{N}_{Truncated}(\mu_\alpha, \sigma_\alpha, 1, 2) \text{ for } m = 1, \dots, M.
\end{aligned} \tag{26}$$

Our experiment investigates the sensitivity to different $C_P$, $C_A$, and $C_D$ realizations. For $C_P$, we power-scale the hierarchical prior of $\alpha$ with a wide range of $\gamma \in [0.1, 10]$, which would have been infeasible with existing methods since they require values close to the mid-range value of no scaling, i.e., $\gamma = 1$ (Kallioinen et al., 2021). To ensure equal amounts of widening and shrinking of the respective distributions during training, we draw the scaling factors from independent uniform distributions in the log space $\gamma \sim \exp(\mathcal{U}(\log(0.1), \log(10)))$. Figure 14a displays the prior predictive distribution of $\alpha$ under different $\gamma$ configurations. We use 100 bootstrap samples on the trial level (i.e., within the observations of each participant) for the data context $C_D$ and describe the details of the approximator context $C_A$ in the following section.

**Neural Network and Training:** All 20 members of the employed deep ensemble constituting $C_A$ are set up and trained independently and identically. Each network uses a hierarchical summary network consisting of two permutation invariant deep set networks (Zaheer et al., 2017) and a standard feedforward network with a softmax output layer as an inference network.

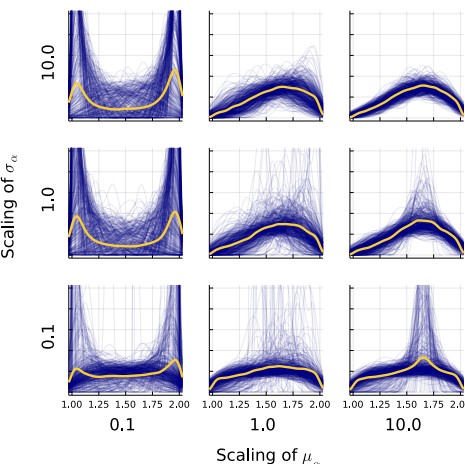

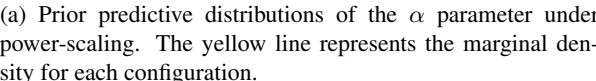

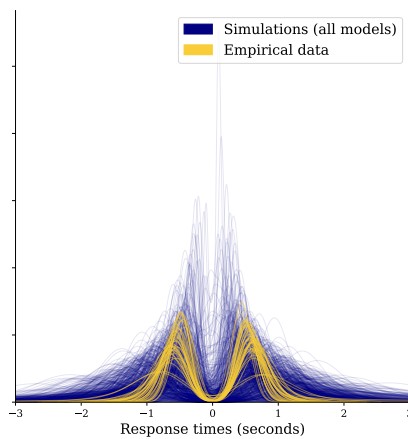

(a) Prior predictive distributions of the $\alpha$ parameter under power-scaling. The yellow line represents the marginal density for each configuration.

(b) Prior pushforward distributions of simulated response times per person contrasted with the empirical distributions per person. Negative response times indicate a decision for the lower decision boundary and positive times for the upper decision boundary.

Figure 14: **Experiment 3**. Prior distributions. a) Prior predictive distributions of the $\alpha$ parameter under maximum widening ($\gamma = 0.1$), no scaling ($\gamma = 1$), and maximum shrinkage ($\gamma = 10$) of the hyperpriors $\mu_\alpha$ and $\sigma_\alpha$. (b) Prior pushforward distributions of the response times simulated by the four compared models (marginalized over $\gamma$) and the empirical response times.

As in Elsemüller et al. (2023), we first pre-train each network on smaller data sets of $40$ simulated participants with $100$ observations each, and afterwards fine-tune on the full data size of $40$ simulated participants with $900$ observations. We use 30 epochs for both phases and an Adam optimizer (Kingma & Ba, 2015) with a cosine decay schedule (initial learning rates of $5 \times 10^{-4}$ for pre-training and $5 \times 10^{-5}$ for fine-tuning).

All computations for this experiment were performed on a single-GPU machine with an NVIDIA RTX 3070 graphics card and an AMD Ryzen 5 5600X processor. Simulating $40\,000$ pre-training and $8\,000$ fine-tuning data sets in the Julia programming language (Bezanson et al., 2017) took 23 minutes. Training the deep ensemble took 21 minutes for pre-training and 45 minutes for fine-tuning per network.

**Additional Results:** Figure 14b displays the prior pushforward distribution of simulated response times and the empirical response times distributions. The informative priors from Elsemüller et al. (2023) assign high densities to the central regions of the empirical distribution. Nevertheless, the results of the typical set approach (Nalisnick et al., 2019; Morningstar et al., 2021) in Figure 15 flag the empirical data as out-of-distribution of each deep ensemble member.

To ensure that the inclusion of $C_P$ in the amortization scope does not lead to a substantially worsened approximation performance, we trained an additional deep ensemble without $C_P$. Table 8 displays validation performance and empirical approximations for the ensemble including $C_P$ and Table 9 for the ensemble without $C_P$. Including $C_P$ does neither lead to a substantial drop in performance nor qualitatively different model comparison results despite power-scaling over a wide range.

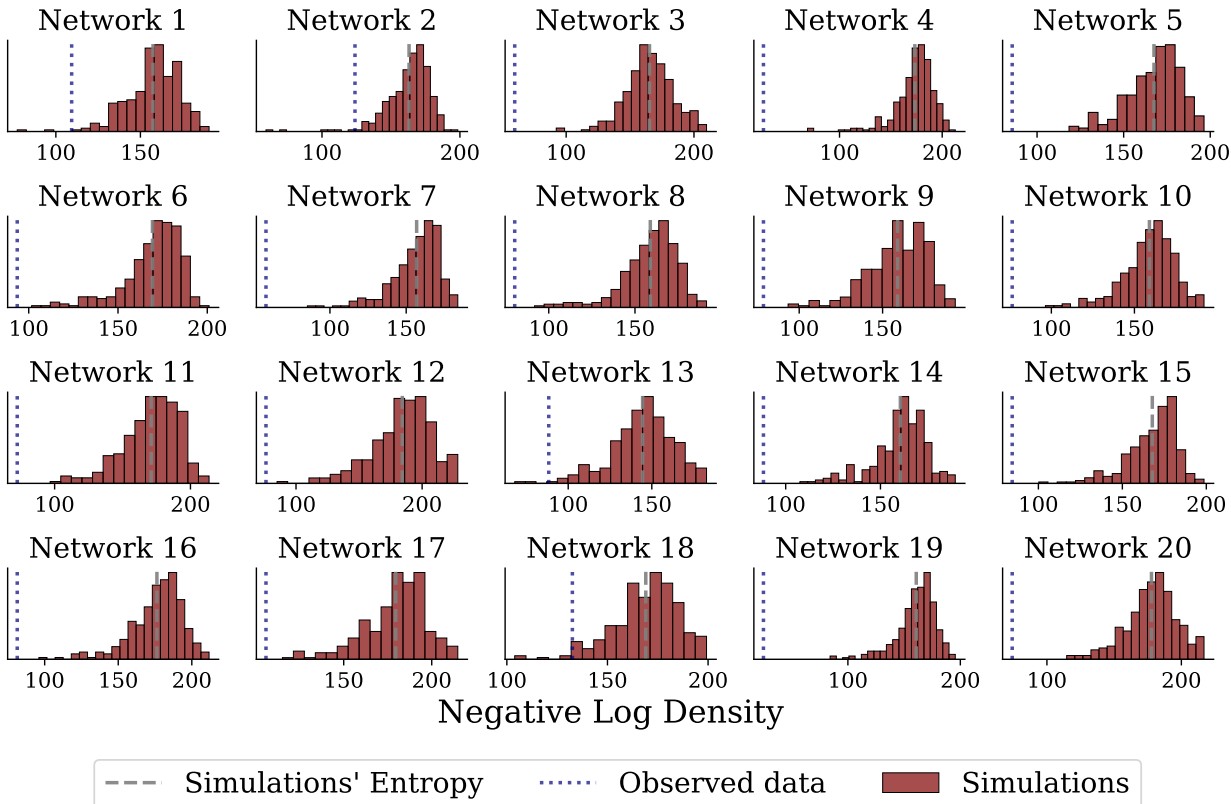

Figure 15: **Experiment 3**. The typical set of learned summary statistics from the model simulations contrasted with the density of the learned summary statistics from the observed data (both distributions are marginalized over $C_P$ but separated for each ensemble member of $C_A$). The empirical data is flagged as out-of-distribution for each ensemble member.

Table 8: **Experiment 3**. SA-ABI validation performance on $8\,000$ held-out simulations and predictions on the empirical data of the deep ensemble trained on power-scaled $C_P$ context from 0.1 to 10.

| | Validation Performance | | | | Empirical Predictions | | | |
|---|---|---|---|---|---|---|---|---|
| | ECE | Brier Score | MAE | Accuracy | $\mathcal{M}_1$ | $\mathcal{M}_2$ | $\mathcal{M}_3$ | $\mathcal{M}_4$ |
| Network 1 | 0.01 | 0.01 | 0.02 | 0.99 | 0.00 | 0.00 | 1.00 | 0.00 |
| Network 2 | 0.01 | 0.03 | 0.05 | 0.96 | 0.00 | 0.00 | 0.92 | 0.08 |
| Network 3 | 0.01 | 0.01 | 0.01 | 0.99 | 0.00 | 0.00 | 0.17 | 0.83 |
| Network 4 | 0.01 | 0.01 | 0.01 | 0.99 | 0.00 | 0.00 | 0.50 | 0.50 |
| Network 5 | 0.01 | 0.01 | 0.01 | 0.99 | 0.00 | 0.00 | 0.96 | 0.04 |
| Network 6 | 0.01 | 0.01 | 0.01 | 0.99 | 0.00 | 0.00 | 0.02 | 0.98 |
| Network 7 | 0.01 | 0.01 | 0.01 | 0.99 | 0.00 | 0.00 | 0.75 | 0.25 |
| Network 8 | 0.01 | 0.01 | 0.02 | 0.99 | 0.00 | 0.00 | 1.00 | 0.00 |
| Network 9 | 0.01 | 0.01 | 0.01 | 0.99 | 0.52 | 0.03 | 0.25 | 0.21 |
| Network 10 | 0.01 | 0.02 | 0.03 | 0.97 | 0.00 | 0.00 | 0.85 | 0.15 |
| Network 11 | 0.01 | 0.01 | 0.01 | 0.99 | 0.00 | 0.00 | 0.37 | 0.63 |
| Network 12 | 0.01 | 0.01 | 0.01 | 0.99 | 0.00 | 0.00 | 0.00 | 1.00 |
| Network 13 | 0.01 | 0.01 | 0.01 | 0.99 | 0.00 | 0.00 | 0.99 | 0.01 |
| Network 14 | 0.01 | 0.02 | 0.03 | 0.98 | 0.00 | 0.00 | 0.99 | 0.01 |
| Network 15 | 0.01 | 0.01 | 0.01 | 0.99 | 0.00 | 0.96 | 0.00 | 0.04 |
| Network 16 | 0.00 | 0.00 | 0.01 | 0.99 | 0.97 | 0.00 | 0.03 | 0.00 |
| Network 17 | 0.01 | 0.02 | 0.02 | 0.98 | 0.00 | 0.00 | 0.62 | 0.38 |
| Network 18 | 0.01 | 0.02 | 0.03 | 0.98 | 0.00 | 0.00 | 0.91 | 0.09 |
| Network 19 | 0.01 | 0.01 | 0.01 | 0.99 | 0.00 | 0.04 | 0.49 | 0.46 |
| Network 20 | 0.01 | 0.01 | 0.01 | 0.99 | 0.00 | 0.00 | 0.08 | 0.92 |
| Average | 0.01 | 0.01 | 0.02 | 0.99 | 0.07 | 0.05 | 0.54 | 0.33 |
| Std. Deviation | 0.00 | 0.01 | 0.01 | 0.01 | 0.24 | 0.21 | 0.40 | 0.36 |

Table 9: **Experiment 3**. ABI validation performance on $8\,000$ held-out simulations and predictions on the empirical data of the deep ensemble trained without $C_P$ context.

| | Validation Performance | | | | Empirical Predictions | | | |
|---|---|---|---|---|---|---|---|---|
| | ECE | Brier Score | MAE | Accuracy | $\mathcal{M}_1$ | $\mathcal{M}_2$ | $\mathcal{M}_3$ | $\mathcal{M}_4$ |
| Network 1 | 0.00 | 0.00 | 0.00 | 1.00 | 0.00 | 0.00 | 0.01 | 0.99 |
| Network 2 | 0.00 | 0.01 | 0.01 | 0.99 | 0.00 | 0.00 | 0.00 | 1.00 |
| Network 3 | 0.01 | 0.01 | 0.02 | 0.98 | 0.00 | 0.00 | 0.99 | 0.01 |
| Network 4 | 0.01 | 0.02 | 0.03 | 0.98 | 0.00 | 0.00 | 0.60 | 0.40 |
| Network 5 | 0.01 | 0.01 | 0.03 | 0.98 | 0.00 | 0.00 | 0.88 | 0.12 |
| Network 6 | 0.00 | 0.00 | 0.00 | 1.00 | 1.00 | 0.00 | 0.00 | 0.00 |
| Network 7 | 0.00 | 0.00 | 0.00 | 1.00 | 0.00 | 0.00 | 0.00 | 1.00 |
| Network 8 | 0.01 | 0.01 | 0.02 | 0.98 | 0.00 | 0.00 | 0.00 | 1.00 |
| Network 9 | 0.00 | 0.00 | 0.00 | 1.00 | 0.00 | 1.00 | 0.00 | 0.00 |
| Network 10 | 0.00 | 0.00 | 0.00 | 1.00 | 0.72 | 0.16 | 0.11 | 0.00 |
| Network 11 | 0.00 | 0.00 | 0.00 | 1.00 | 0.01 | 0.42 | 0.00 | 0.56 |
| Network 12 | 0.01 | 0.02 | 0.04 | 0.97 | 0.00 | 0.00 | 0.91 | 0.09 |
| Network 13 | 0.01 | 0.01 | 0.01 | 0.99 | 0.00 | 0.00 | 0.00 | 1.00 |
| Network 14 | 0.00 | 0.00 | 0.00 | 1.00 | 0.00 | 0.00 | 1.00 | 0.00 |
| Network 15 | 0.01 | 0.01 | 0.01 | 0.99 | 0.00 | 0.00 | 1.00 | 0.00 |
| Network 16 | 0.00 | 0.00 | 0.00 | 1.00 | 0.00 | 0.00 | 0.98 | 0.02 |
| Network 17 | 0.00 | 0.00 | 0.00 | 1.00 | 0.00 | 0.25 | 0.01 | 0.74 |
| Network 18 | 0.00 | 0.00 | 0.00 | 1.00 | 0.00 | 0.00 | 0.86 | 0.14 |
| Network 19 | 0.01 | 0.01 | 0.01 | 0.99 | 0.00 | 0.00 | 0.00 | 1.00 |
| Network 20 | 0.01 | 0.02 | 0.04 | 0.97 | 0.00 | 0.00 | 0.88 | 0.11 |
| Average | 0.00 | 0.01 | 0.01 | 0.99 | 0.09 | 0.09 | 0.41 | 0.41 |
| Std. Deviation | 0.01 | 0.01 | 0.01 | 0.01 | 0.27 | 0.24 | 0.46 | 0.44 |

