# OpenReview forum: "Sensitivity-Aware Amortized Bayesian Inference"
_TMLR — Accepted by TMLR_

### Review · Reviewer_dZmd · 2024-06-02

**Summary Of Contributions:**

This paper proposes a fully amortized approach for facilitating sensitivity analysis in simulation-based inference (SBI), an important but somewhat overlooked topic in SBI. The key idea is to treat the various sensitivity sources (e.g. different assumption of the underlying data generating process used in simulation, different choices of prior used in simulation, etc.) as context variables. These context variables serve as additional inputs to the neural network in amortized Bayesian inference, adding an outer level of amortization. Upon convergence, one can either quantitatively or qualitatively inspect the sensitivity w.r.t specific sources by varying the values of the corresponding input context variables, thereby enabling easy sensitivity analysis. Experiments on three real-world models in epidemics, climate science and cognitive science demonstrate the potential of the method.

**Audience:**

Yes

**Claims And Evidence:**

Yes

**Requested Changes:**

- Reproducibility: It is encouraged that upon acceptance, the author will provide the code of the method and the simulation models for reproducible research;
- Presentation: It may also be helpful to group the topics in sec. 3 into two subgroups for further improving (the already very good) clarity. Furthermore, it *might* also be useful to provide a table summarizing different context variables considered in this work along with corresponding real-world examples.

**Strengths And Weaknesses:**

**Strengths**

- The problem studied in this work (i.e. sensitivity analysis for SBI) is important and is of great interest within SBI community. The reviewer himself had also considered the same research question for a long time;
- The method proposed is simple and elegant, with necessary implementation details well-explained;
- The paper is mostly well-written and easy to follow. I personally enjoy reading the paper very much;
- The proposed method is verified with several real-world examples. In addition, the author also include discussions on the practical implications found by their amortized sensitivity-analysis method (see e.g. the experiments on climate modeling), which is insightful.

**Weaknesses**

The paper targets at an overlooked problem in SBI and proposes an elegant approach to fix it, with good empirical results. I don't really have too much to complain about. The only comments are:

- Reproducibility: despite the important method developed and the interesting results, there seem to be no way to access the code of the proposed method nor the code of the simulator, which is a bit disappointing;

- Explicit comparison to non-amortized sensitive analysis approach. The work may benefit from an explicit discussion on how their amortize method supports sensitive analysis that is not possible with non-amortized approach. This could be done by either highlighting in the methodology section or in the experiment.

---

### Review · Reviewer_6Cay · 2024-06-05

**Summary Of Contributions:**

This paper introduces a method to efficiently integrate sensitivity analysis into Amortized Bayesian inference by using implicit context variables to represent different sources of perturbations. The resulting approach, called Sensitivity-Aware Amortized Bayesian Inference (SA-ABI), provides insights into the sensitivity of inference concerning the likelihood, prior, approximator, and data perturbations without the need to re-perform the entire analysis for each instance of the context variable. It is versatile, applicable to different types of Bayesian inference, including parameter estimation and model comparison, and is suitable for large-scale applications. The effectiveness of this methodology is demonstrated through real-world applications, showcasing its practical utility.

**Audience:**

Yes

**Broader Impact Concerns:**

I don't see any broader impact concerns

**Claims And Evidence:**

Yes

**Requested Changes:**

- [crucial] Clarification with respect to my points above throughout the paper on the type of approximator sensitivity addressed and its connection to simulation gap/ model misspecification.
- [crucial] Figure 4 can benefit from a more detailed explanation which I think will help with the issues mentioned above. For example, the text reads ` Strikingly, Figure 4 reveals highly inconsistent predictions of the ensemble members '. Is this conclusion based on the high uncertainty of the posteriors? Does that directly link to high disparity between ensemble members or could it be due to other factors? A further explanation regarding the last two subplots would also be helpful to the reader, especially as $C_A$, $C_D$ and $C_P$ are all incorporated in different ways. For example, for the third subplot how do the illustrated posteriors take into account all the different data perturbations? A figure like 2b or 3a for at least a subset of different $C_D$ instances would be helpful here.
- [non-crucial] In equation 8 how is the prior over $C_T$ normally chosen? Is it something simple like a uniform prior over different realisations of the context variable?
- [non-crucial] Typo: above equation 6 in the two brackets I think section 3.3 should be 3.4 and section 3.4 should be 3.5

**Strengths And Weaknesses:**

Strengths:

I thoroughly enjoyed the paper and believe it will be of great interest to the TMLR audience, as well as the wider SBI/ABI community. It offers an easily applied and computationally efficient method to incorporate sensitivity analysis in ABI, which is especially important for neural-based methods. The approach is versatile enough to handle different instances of context variables, as well as allowing users to easily evaluate sensitivity to new context instances of likelihood and prior at inference time. Experiments 1 and 2 provide great demonstrations of the method's effectiveness in sensitivity analysis of prior and likelihood choices.

Weaknesses:

The main limitation I identified is the lack of clarity with respect to the approximator sensitivity and its relationship to the issue of model misspecification and simulation gaps. Firstly, the authors later differentiate between closed-world and open-world settings. In the closed-world setting, which in this case refers to the setting where the data at inference time is sampled from the simulator hence we are certain that the model is well-specified, I think approximation sensitivity is clear and performance variability between ensemble members can give insights on the sensitivity of the neural approximator e.g. due to finite training or suboptimal convergence, as mentioned in the paper.

However, in the open-world setting things are less clear. The paper refers to detecting approximator sensitivity as a way to get insights on the effect of simulation gaps due to a misspecified simulator relative to the data-generating process (this is mentioned in the abstract where approximator sensitivity refers to ‘unreliable approximation due to e.g. model misspecification' and later in Section 3.4 where approximator sensitivity is defined as `the variability of inferential results due to the approximation method employed'. ). It further hypothesizes that 'variability across the M ensemble members in the open world despite consistent performance in the closed world indicates a simulation gap'. There are two issues that need clarification: 1) In practice does this suggest that one should perform inference on simulated data and on observed data (potentially from a different model) to assess such variability? Is this also done in Experiment 3? and more importantly 2) It is generally not clear throughout the paper how approximator sensitivity is related to simulation gaps and how variability across performances of ensemble members indicate model misspecification. To explain myself further:

- The above hypothesis is motivated by relevant works on the robustness of SBI methods in misspecified settings, such as Cannon et al. (2022), who highlight that in misspecified settings, the main issue affecting the performance of neural SBI is poor performance in OOD settings, such as in the presence of simulation gaps. Sensitivity to such gaps usually stems from the misspecified simulator and the poor generalization capabilities of the neural approximator, rather than the quality of the posterior approximation itself. In other words, the approximator might be accurately approximating the "target" posterior (i.e., the Bayesian posterior based on simulated data), but that posterior might not be the "optimal" posterior with respect to the data-generating process. This issue arises because the Bayesian posterior itself (which the approximation targets) is not robust to model misspecification. This raises the question of whether the approximation sensitivity refers to sensitivity to the quality of approximation of the Bayesian posterior as in the well-specified setting or to the sensitivity of the resulting inference due to model misspecification as in Cannon et al. 2022 and other citations in the paper?

- The above point also leads to the question of how such sensitivity is detected. In the paper an ensemble of neural networks (of the same configuration) is used, and sensitivity is defined as the variation in inference results among the ensemble members. It is not entirely clear how variability in the performance of ensemble members would quantify sensitivity to model misspecification or data contamination. I can imagine a scenario where all ensemble members perform similarly but lead to misleading inference outcomes because they are all based on the same misspecified simulator.

---

### Review · Reviewer_ME2Q · 2024-06-24

**Summary Of Contributions:**

This paper introduces a method for using amortised Bayesian inference for sensitivity analysis. This is done by
training deep ensemble networks which allow models and data to be readily changed and quickly obtaining a new
posterior. The resulting posteriors seem no worse than what is obtained with normal amortized inference.

**Audience:**

Yes

**Broader Impact Concerns:**

No concerns about ethical implications of this work

**Claims And Evidence:**

Yes

**Requested Changes:**

More details on what were the models used for the experiments. It would also help to have an example which uses
the scoring rules just to show them in action.

**Strengths And Weaknesses:**

Strengths:

This is highly original work that has the potential to be really significant for evaluating and
exploring models using amortised inference. The experiments are well-designed and effective at
demonstrating how sensitivity to the prior, data, and approximator looks like in several real-world
examples. The writing is very clear and straightforward to follow. This work should definitely be published.

Weaknesses:

Some of the results can be hard to interpret. For example for Figure 4, it's not clear what are these
four models and how do they differ. I couldn't find detailed answers even in the appendix. Also although
scoring rules are mentioned, it doesn't seem any are used for the experiments. It would make the paper
stronger if those can be used in some experiments. It would also be helpful to show that results are
still calibrated for all context variables.

---

> ### Author Response · Authors · 2024-06-26
> **Response to Reviewer ME2Q**
>
> Thank you for your thoughtful review of our paper. We appreciate your assessment of the originality and potential impact of our work as well as your recommendations for further improvements. Below, we address your comments and describe the revisions we have made.
>
> ## Clarifications on Experiment 3
>
> Thank you for drawing our attention to this issue. We agree that the appendix was not precise enough when it comes to the specific models that we compare in our experiment. We added the following list to the “Model Setup” section of the Appendix on Experiment 3 to clarify the components in each model:
>
> “Therefore, we compare the following four models in this experiment:
> - Basic DDM ($\mathcal{M}_1$): The classic four-parameter formulation with the model parameters $v$, $t_0$, $a$, and $z_r$.
> - Basic LFM ($\mathcal{M}_2$): Equals the basic DDM plus the additional stability parameter $\alpha$ controlling the tail behavior of the noise distribution.
> - Full DDM ($\mathcal{M}_3$): Equals the basic DDM plus inter-trial variability parameters $s_v$, $s_t$, and $s_z$.
> - Full LFM ($\mathcal{M}_4$): Equals the full DDM plus the additional stability parameter $\alpha$.”
>
> We appreciate additional suggestions to improve the clarity of our model descriptions.
>
> ## Usage of Scoring Rules
>
> We found that the standard Kullback-Leibler (KL) divergence for parameter estimation (i.e., cross-entropy between pdfs) and model comparison (i.e., cross-entropy between pmfs) modified to include context variables (Equations 6 and 7) is well-suited as a scoring rule for the goals of our experiments. As our training modifications simply expand the conditioning space during training, we believe that our results may generalize well to any other scoring rules used in SBI. Nevertheless, we agree that concrete examples of such other scoring rules and when they are used would improve this part of our work. We edited the beginning of the scoring rule paragraph in Section 3.2 accordingly:
>
> “Our approach seamlessly generalizes to other strictly proper losses (Gneiting & Raftery, 2007) which can be used as training objectives for amortized inference (Pacchiardi & Dutta, 2021). **For example, Schmitt et al. (2023) use a modified loss to enable density–based misspecification detection in parameter estimation, while Jeffrey & Wandelt (2024) propose losses to directly optimize for Bayes factors in model comparison problems**.” (highlighted text is new)
>
> Given the excellent performance of the KL losses in our experiments, we would delegate a systematic comparison of different scoring rules for various benchmark models and simulation budgets to future empirical work. Accordingly, we have included an explicit discussion of this potential research avenue to the Conclusion section. However, we are also open to suggestions for concrete comparisons if the reviewer holds such a comparison absolutely crucial. Lastly, does your comment concerning calibration refer to the behavior under different scoring rules or our current setting? We would be happy to clarify this further.
>
>  ## References:
>
> Schmitt, M., Bürkner, P. C., Köthe, U., & Radev, S. T. (2023). Detecting model misspecification in amortized Bayesian inference with neural networks. DAGM German Conference on Pattern Recognition.
>
> Jeffrey, N., & Wandelt, B. D. (2024). Evidence Networks: simple losses for fast, amortized, neural Bayesian model comparison. Machine Learning: Science and Technology, 5(1), 015008.

---

### Author Response · Authors · 2024-08-14
**Concluding responses (Part 1/2)**

We thank all reviewers and the action editor for the high quality of the review process, taking the time to carefully evaluate our work, and providing highly constructive feedback. We address all remaining comments below.

# Action Editor
## Limits of Amortization
 We fully agree that the additional complexity induced by amortizing over families of likelihood/prior distributions depends on the diversity of the resulting generative patterns. Quantifying this relationship numerically is challenging, since the diversity of the outcomes depends on assumptions about the generative model and the prior and/or likelihood variations, with the resulting increase in complexity additionally influenced by the architecture of the neural approximator. We revised the discussion of amortization limitations in the Methods section to make the general relationship clearer and provide recommendations for practitioners based on our empirical findings:

“A natural question that immediately arises is *whether the resulting sensitivity-aware posterior is less accurate than the corresponding fixed-context posterior*. Intuitively, the answer depends on the sampling diversity of the contextualized joint model $p(\boldsymbol{x}, \boldsymbol{y}, C_T)$ and the potential for reaping the benefits of weight sharing: If the associated likelihood and prior variations instantiate generative models with wildly different behaviors, weight sharing may not be advantageous, resulting in diminishing returns from amortized training over the context variables $C_T$. Fortunately, the set of plausible choices for a given modeling problem typically leads to similar generative patterns, so that weight sharing is much more efficient than separate approximation. Indeed, our experiments demonstrate this for several representative model families even under small simulation budgets. Nevertheless, if amortization over very different simulators is desired, we recommend increasing the expressiveness of the neural approximator, the simulation budget, and the allotted training time.”

## p2 "hundred thousand model configurations"
We clarified this part of the contributions by stating the exact number of maximum configurations we tested at inference time:

“We demonstrate the utility of SA-ABI for Bayesian parameter estimation as well as Bayesian model comparison in three real-world scenarios of scientific interest, investigating sensitivity under up to $162\,000$ configurations.”

Since the number of maximum configurations is the product of prior, approximator, and data variations in Experiment 3, we found that specifying the details of the prior variations in this particular experiment seems misplaced in the Introduction. We instead added more explanations directly in Experiment 3:

 “The complexity of amortizing over the prior space is balanced by two aspects: While the scaling only affects the hyperpriors of the additional $\alpha$ parameter in $\mathcal{M}_2$ and $\mathcal{M}_4$, scaling up to a factor of $10$ leads to much more extreme variations than the usual perturbations in likelihood-based settings of up to a factor of $2$ (Kallioinen et al., 2021).”

## Additional References
Thanks for pointing us to the additional relevant references! We included all references, except reference [1] mentioned by reviewer dZmd. It does not cover end-to-end learnable summary statistics but rather a specific method for constructing summary statistics for ABC and we couldn’t seamlessly integrate it.

---

> ### Author Response · Authors · 2024-08-14
> **Concluding responses (Part 2/2)**
>
> ## Sensitivity to the Choice of Neural Network Architecture
> We believe that sensitivity to the neural network architecture can be understood as a form of approximator sensitivity in our framework and addressed with hyperparameter ensembles. Accordingly, we added some examples to our discussion of hyperparameter ensembles to highlight the connection to SBI:
>
> “Hyperparameters that are particularly relevant in SBI include the architecture of the summary network (e.g., inductive bias induced by the architecture, number of learned summary statistics), the choice of inference network (e.g., architecture, number of trainable weights), and common hyperparameters that are ubiquitous in deep learning (e.g., learning rate, dropout).”
>
> Concerning our specific architectural choices, we followed prior work whenever available for all experiments. For example, Experiment 1 chooses a similar summary and inference network architecture as Radev et al. (2021) with converging results, Experiment 2 follows the summary network architecture from Diffenbaugh & Barnes (2023), and Experiment 3 follows the summary and inference network architecture from Elsemüller et al. (2023). We updated the introduction of the Experiments section to now clearly state that the network architectures are described in the appendix of each experiment. Since we use the same architecture for comparing between ABI and SA-ABI and additionally obtain converging results between different experiments with different architectures, we believe that our results are not sensitive to a specific architecture.
>
> # Reviewer dZmd
> ## Communication of Scaling Advantages
> We appreciate your suggestions regarding the communication of the scaling advantages. We think that our current comparison with ABI is the most fair and challenging benchmark, since obtaining approximations for e.g., the 1000 test data sets in Experiment 1 (not considering any further sensitivity analyses) would already be a major challenge for non-amortized methods.
>
> ## Structure of the Methods Section
> The final version features the suggested structure, which we think greatly improves the clarity of the Methods section.
>
> ## Sensitivity to the Choice of Summary Statistics
> Thank you for raising this interesting point! Since our work focuses on end-to-end learnable summary statistics, we included different choices for the summary network as potential approximator sensitivity sources in our response to the action editor above. Manual expert statistics obtained during preprocessing would fall under data sensitivity, whereas summary statistics that are learned upfront (i.e., decoupled from the posterior approximator) might be considered as an edge case that could fall into either category.

---

### Decision · Action_Editor_SRiC · 2024-07-29

**Recommendation:** Accept with minor revision

**Comment:**

All reviewers mention that they enjoyed reading the paper. Some reviewers had some concerns about the clarity of some aspects of the methodology but this was cleared up in the authors' response.

I concur with the reviewers that this is a good paper that should be accepted. I do, however, have the following comments that I would like the authors to take into account in their final version:

- p2: "hundred thousand model configurations" Please clarify how distinct the different configurations are. For example, I imagine that in a hypothetical setup where we specify a prior on a mean, choosing a Gaussian and varying its mean results in scenarios that easier to amortise over than choosing different prior families (e.g. Bernoulli, spike-and-slab prior, and Poisson distribution) that results in more different prior predictive outputs.

- Section 2.2: Regarding the Cranmer et al, 2020 citation for implicit likelihoods. Please clarify that this is just one possible reference, e.g. by adding a "e.g.". The earliest reference to my knowledge is https://www.jstor.org/stable/2345504 Another earlier review paper that discuss this are https://link.springer.com/article/10.1007/s11222-011-9288-2

- I think references [1] and [3] mentioned by Reviewer dZmd should be added to section 2.4. Reference https://openreview.net/forum?id=SRDuJssQud would also be a good addition to this section.

- [Important] Please add a discussion on the limits of the amortisation, in terms of the diversity of the different scenarios and scalability. I suppose as the number of the different scenarios and their diversity increases, the simulation budget needs to increase as well, with a rate depending on the similarity of the scenarios. Can you quantify how your results change as the diversity of different models considered increases?

- Please further add a discussion on the sensitivity of the setup to the choice of the neural network architecture that you use, and how you select them.

**Audience:**

This paper is about amortised Bayesian inference for sensitivity analysis in the context of simulator models. The paper will be of interest to the TMLR audience, in particular to the sub-community interested in approximate Bayesian inference.

**Claims And Evidence:**

The reviewers all agree that the claims are accurate and convincing. I concur.